# Discovering Symbolic Differential Equations with Symmetry Invariants

**Jianke Yang** *jiy065@ucsd.edu*
*University of California, San Diego*

**Manu Bhat** *mbhat@ucsd.edu*
*University of California, San Diego*

**Bryan Hu** *brhu@ucsd.edu*
*University of California, San Diego*

**Yadi Cao** *yac066@ucsd.edu*
*University of California, San Diego*

**Nima Dehmamy** *nima.dehmamy@ibm.com*
*IBM Research*

**Robin Walters** *r.walters@northeastern.edu*
*Northeastern University*

**Rose Yu** *roseyu@ucsd.edu*
*University of California, San Diego*

**Reviewed on OpenReview:** *https://openreview.net/forum?id=9t1dEyYfPc*

## Abstract

Discovering symbolic differential equations from data uncovers fundamental dynamical laws underlying complex systems. However, existing methods often struggle with the vast search space of equations and may produce equations that violate known physical laws. In this work, we address these problems by introducing the concept of *symmetry invariants* in equation discovery. We leverage the fact that differential equations admitting a symmetry group can be expressed in terms of differential invariants of symmetry transformations. Thus, we propose using these invariants as atomic entities in equation discovery, ensuring the discovered equations satisfy the specified symmetry. Our approach integrates seamlessly with existing equation discovery methods such as sparse regression and genetic programming, improving their accuracy and efficiency. We validate the proposed method through applications to various physical systems, such as Darcy flow and reaction-diffusion, demonstrating its ability to recover parsimonious and interpretable equations that respect the laws of physics.

## 1 Introduction

Differential equations describe relationships between functions representing physical quantities and their derivatives. They are crucial in modeling a wide range of phenomena, from fluid dynamics and electromagnetic fields to chemical reactions and biological processes, as they succinctly capture the underlying principles governing the behavior of complex systems. The discovery of governing equations in symbolic forms from observational data bridges the gap between raw data and fundamental understanding of physical systems. Unlike black-box machine learning models, symbolic equations provide interpretable insights into the structure and dynamics of the systems of interest. In this paper, we aim to discover symbolic partial differential

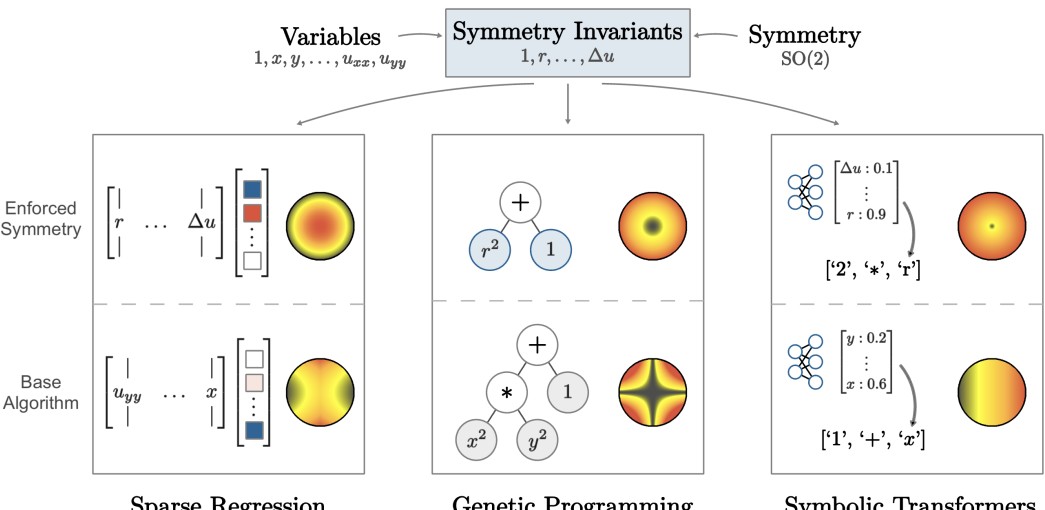

Figure 1: Our framework enforces symmetry in equation discovery by using symmetry invariants. We highlight three discovery algorithms in their original form (bottom row) and when constrained to only use symmetry invariants (top row). The colored circles visualize the predicted functions on a circular domain and demonstrate that using symmetry invariants guarantees a symmetric output.

equations (PDEs) in the form

$$F(\mathbf{x}, u^{(n)}) = 0, \tag{1}$$

where $\mathbf{x}$ denotes the independent variables, $u^{(n)}$ consists of the dependent variable $u$ and all of its up-to-$n$th order partial derivatives.

While it has long been an exclusive task for human experts to identify governing equations, symbolic regression (SR) has emerged as an increasingly popular approach to automate the discovery.[1] SR constructs expressions from a predefined set of atomic entities, such as variables, constants, and mathematical operators, and fits the expressions to data by numerical optimization. Common methods include sparse regression (Brunton et al., 2016; Champion et al., 2019), genetic programming (Cranmer et al., 2019; 2020; Cranmer, 2023), neural networks (Kamienny et al., 2022), etc.

However, symbolic regression algorithms may fail due to the vastness of the search space or produce more complex, less interpretable equations that overfit the data. A widely adopted remedy to these challenges is to incorporate inductive biases derived from physical laws, such as symmetry and conserved quantities, into equation discovery algorithms. Implementing these physical constraints narrows the space for equations and expedites the search process, and it also rules out physically invalid or unnecessarily complex equations.

Among the various physical constraints, symmetry plays a fundamental role in physical systems, governing their invariances under transformations such as rotations, translations, and scaling. Previous research has shown the benefit of incorporating symmetry in equation discovery, such as reducing the dimensionality of the search space and promoting parsimony in the discovered equation (Yang et al., 2024). However, the scopes of existing works exploiting symmetry are limited in terms of the types of equations they can handle, the compatible base algorithms, etc. For example, Udrescu & Tegmark (2020) deals with algebraic equations; Otto et al. (2023) deals with ODE systems; Yang et al. (2024) applies to sparse regression but not other SR algorithms.

In this paper, we propose a general procedure based on *symmetry invariants* to enforce the inductive bias of symmetry with minimal restrictions in the types of equations and SR algorithms. Specifically, we leverage the fact that a differential equation can be written in terms of the invariants of symmetry transformations if it admits a certain symmetry group. Thus, instead of operating on the original variables, our method uses the symmetry invariants as the atomic entities in symbolic regression, as depicted in Figure 1. These

---

[1]While some literature uses *symbolic regression* specifically for GP-based methods, we use the term interchangeably with *equation discovery* to refer to all algorithms for learning symbolic equations.

invariants encapsulate the essential information while automatically satisfying the symmetry constraints. Consequently, the discovered equations are guaranteed to preserve the specified symmetry. In summary, our main contributions are listed as follows:

- We propose a general framework to enforce symmetry in differential equation discovery based on the theory of differential invariants.

- Our approach can be easily integrated with existing symbolic regression methods, such as sparse regression and genetic programming, and improves their accuracy and efficiency for differential equation discovery.

- We show that our symmetry-based approach is robust in challenging setups in equation discovery, such as noisy data and imperfect symmetry.

**Notations.** Throughout the paper, subscripts are usually reserved for partial derivatives, e.g. $u_t := \partial u / \partial t$, and $u_{xx} := \partial^2 u / \partial x^2$. Superscripts are used for indexing vector components or list elements. We use Einstein notation, where repeated indices are summed over. Matrices, vectors and scalars are denoted by capital, bold and regular letters, respectively, e.g. $W, \mathbf{w}, w$. These conventions may admit exceptions for clarity or context. See Table 2 for a full description of notations.

## 2 Related Works

**Symbolic Regression.** Given the dataset $\{(x^i, y^i)\} \subset X \times Y$, symbolic regression (SR) aims to model the function $y = f(x)$ by a symbolic equation. A popular method for symbolic regression is genetic programming (GP) (Schmidt & Lipson, 2009; Gaucel et al., 2014), which leverages evolutionary algorithms to explore the space of possible equations and has demonstrated success in uncovering governing laws in various scientific domains such as material science (Wang et al., 2019), climate modeling (Grundner et al., 2023), cosmology (Cranmer et al., 2020), etc. Various software have been developed for GP-based symbolic regression, e.g. Eureqa (Dubčáková, 2011) and PySR (Cranmer, 2023).

Another class of methods is sparse regression (Brunton et al., 2016), which assumes the function to be discovered can be written as a linear combination of predefined candidate functions and solves for the coefficient matrix. It has also been extended to discover more general equations, such as equations in latent variables (Champion et al., 2019) and PDEs (Rudy et al., 2017).

Neural networks have also shown their potential in symbolic regression. Martius & Lampert (2016); Sahoo et al. (2018) represents a few earliest attempts, where they replace the activation functions in fully connected networks with math operators and functions, so the network itself translates to a symbolic formula. Other works represent mathematical expressions as sequences of tokens and train neural networks to predict the sequence given a dataset of input-output pairs. For example, Petersen et al. (2019) trains an RNN with policy gradients to minimize the regression error. Biggio et al. (2021), Kamienny et al. (2022) and Holt et al. (2023) pre-train an encoder-decoder network over a large amount of procedurally generated equations and query the pretrained model on a new dataset of input-output pairs at test time.

The aforementioned symbolic regression methods can be improved by incorporating specific domain knowledge. For example, AI Feynman (Udrescu & Tegmark, 2020; Udrescu et al., 2020) uses properties like separability and compositionality to simplify the data. Cranmer et al. (2020) specifies the overall skeleton of the equation and fits each part with genetic programming independently. The goal of this paper falls into this category – to use the knowledge of symmetry to reduce the search space of symbolic regression and improve its accuracy and efficiency.

Recently, Large Language Models (LLMs) have emerged as an alternative for SR, using pre-trained scientific priors to propose sequential hypothesis (Merler et al., 2024) or to guide genetic programming (Shojaee et al., 2024), balancing the efficiency of domain knowledge with the robustness of evolutionary search. However, current LLM-based methods often rely on memorizing known equations rather than facilitating genuine discovery, and their guidance lacks interpretability, specifically, the reasoning behind their suggestions,

evidenced by a recent benchmark specially designed for LLM-SR (Shojaee et al., 2025). A recent effort sought to improve interpretability by binding symbolic evolution with natural language explanations (Grayeli et al., 2024). However, this method relies on frontier LLMs to conduct the evolution of the natural language components, rendering the process itself opaque. These limitations highlight the need for approaches that enhance the controllability and explainability of the prior knowledge injected, ensuring more transparent and trustworthy discovery.

**Discovering Differential Equations.** While it remains in the scope of symbolic regression, the discovery of differential equations poses additional challenges because the derivatives are not directly observed from data. Building upon the aforementioned SINDy sparse regression (Brunton et al., 2016), Messenger & Bortz (2021a;b) formulates an alternative optimization problem based on the variational form of differential equations and bypasses the need for derivative estimation. A similar variational approach is also applied to genetic programming (Qian et al., 2022). Various other improvements have been made, including refined training procedure (Rao et al., 2022), relaxed assumptions about the form of the equation (Kaheman et al., 2020), and the incorporation of physical priors (Xie et al., 2022; Bakarji et al., 2022; Lee et al., 2022; Messenger et al., 2024).

**PDE Learning and Surrogate Modeling.** Beyond symbolic regression, there is a broad line of work on learning PDEs and their solution operators directly from data using non-symbolic surrogates. Neural operator methods aim to approximate nonlinear operators mapping initial or boundary data to PDE solutions, providing highly expressive black-box solvers for families of PDEs (Lu et al., 2021; Li et al., 2021). Gaussian processes and kernel methods have also been developed to solve nonlinear PDEs and related inverse problems (Chen et al., 2021). A complementary body of work focuses on learning effective coarse-grained dynamics or closures from fine-scale simulations and spatiotemporal data, including reservoir-computing and recurrent architectures (Vlachas et al., 2020), machine-learned coarse-scale PDEs from microscopic data (Lee et al., 2020), numerical bifurcation analysis and intrinsic-coordinate models inferred from simulators (Galaris et al., 2022; Floryan & Graham, 2022), and multiscale frameworks that learn reduced stochastic or PDE models for complex systems (Vlachas et al., 2022; Lee et al., 2023; Dietrich et al., 2023; Fabiani et al., 2024). These approaches typically prioritize predictive accuracy and efficient surrogate modeling over known symbolic structure. In contrast, our framework, along with other SR methods, seeks to recover interpretable closed-form PDEs whose terms can be inspected and analyzed, while still leveraging data-driven tools.

**PDE Symmetry in Machine Learning.** Symmetry is an important inductive bias in machine learning. In the context of learning differential equation systems, many works encourage symmetry in their models through data augmentation (Brandstetter et al., 2022), regularization terms (Akhound-Sadegh et al., 2023; Zhang et al., 2023; Dalton et al., 2024), and self-supervised learning (Mialon et al., 2023). Strictly enforcing symmetry is also possible, but is often restricted to specific symmetries and systems (Wang et al., 2021; Gurevich et al., 2024). For more general symmetries and physical systems, enforcing symmetry often requires additional assumptions on the form of equations, such as the linear combination form in sparse regression (Otto et al., 2023; Yang et al., 2024). To the best of our knowledge, our work is the first attempt to strictly enforce general symmetries of differential equations for general symbolic regression methods. A more detailed discussion of the connections and differences between our work and other symmetry-based equation discovery methods is provided in Appendix D.

## 3 Background

### 3.1 PDE Symmetry

This section introduces the basic concepts of partial differential equations and their symmetry. For a more thorough understanding of Lie point symmetry of PDEs, we refer the readers to Olver (1993).

**Partial Differential Equations.** We consider PDEs in the form $F(\mathbf{x}, u^{(n)}) = 0$, as given in (1). We restrict ourselves to a single equation and a single dependent variable here, though generalization to multiple equations and dependent variables is possible. We use $\mathbf{x} \in X \subset \mathbb{R}^p$ to denote all independent variables. For example,

$\mathbf{x} = (t, x)$ for a system evolving in 1D space. Note that the bold $\mathbf{x}$ refers to the collection of all independent variables while the regular $x$ denotes the spatial variable. Then, $u = u(\mathbf{x}) \in U \subset \mathbb{R}$ is the dependent variable; $u^{(n)} = (u, u_x, ...)$ denotes all up to $n$th-order partial derivatives of $u$; $(\mathbf{x}, u^{(n)}) \in M^{(n)} \subset X \times U^{(n)}$, where $M^{(n)}$ is the $n$th order **jet space** of the total space $X \times U$. $M^{(n)}$ and $u^{(n)}$ are also known as the $n$th-order **prolongation** of $X \times U$ and $u$, respectively.

**Symmetry of a PDE.**  A point symmetry $g$ is a local diffeomorphism on the total space $E = X \times U$:

$$g \cdot (\mathbf{x}, u) = (\tilde{\mathbf{x}}(\mathbf{x}, u), \tilde{u}(\mathbf{x}, u)), \tag{2}$$

where $\tilde{\mathbf{x}}$ and $\tilde{u}$ are functions on $E$. The action of $g$ on the function $u(\mathbf{x})$ is induced from (2) by applying it to the graph of $u : X \to U$. Specifically, denote the domain of $u$ as $\Omega \subset X$ and its graph as $\Gamma_u = \{(\mathbf{x}, u(\mathbf{x})) : \mathbf{x} \in \Omega\}$. The group element $g$ transforms the graph $\Gamma_u$ as $\tilde{\Gamma}_u := g \cdot \Gamma_u = \{(\tilde{\mathbf{x}}, \tilde{u}) = g \cdot (\mathbf{x}, u) : (\mathbf{x}, u) \in \Gamma_u\}$.

Since $g$ transforms both independent and dependent variables, $\tilde{\Gamma}_u$ does not necessarily correspond to the graph of any single-valued function. Nevertheless, by suitably shrinking the domain $\Omega$, we can ensure that the transformations close to the identity transform $\Gamma_u$ to the graph of another function. This function with the transformed graph $\tilde{\Gamma}_u$ is then defined to be the transformed function of the original solution $u$, i.e. $g \cdot u = \tilde{u}$ s.t. $\Gamma_{\tilde{u}} = \tilde{\Gamma}_u$. The symmetry of the PDE (1) is then defined:

**Definition 3.1.** A symmetry group of $F(\mathbf{x}, u^{(n)}) = 0$ is a local group of transformations $G$ acting on an open subset of the total space $X \times U$ such that, for any solution $u$ to $F = 0$ and any $g \in G$, the function $\tilde{u} = (g \cdot u)(\mathbf{x})$ is also a solution of $F = 0$ wherever it is defined.

**Infinitesimal Generators.**  Often, the symmetry group of a PDE is a continuous Lie group. In practice, one needs to compute with infinitesimal generators of continuous symmetries, i.e., vector fields. In more detail, we will write vector fields $\mathbf{v} : E \to TE$ on $E = X \times U$ as

$$\mathbf{v} = \xi^j(\mathbf{x}, u)\frac{\partial}{\partial x^j} + \phi(\mathbf{x}, u)\frac{\partial}{\partial u}. \tag{3}$$

Any such vector field generates a one-parameter group of symmetries of the total space $\{\exp(\epsilon\mathbf{v}) : \epsilon \in \mathbb{R}\}$. The symmetries arising from the exponentiation of a vector field moves a point in the total space along the directions given by the vector field. We will specify symmetries by vector fields in the following sections. For instance, $\mathbf{v} = x\partial_y - y\partial_x$ represents the rotation in $(x, y)$-plane; $\mathbf{v} = \partial_t$ corresponds to time translation.

To analyze the symmetry of PDEs, we must know how it transforms not only the variables, but also their derivatives accordingly. The group transformations on derivatives are formalized by **prolonged** group actions and infinitesimal actions on the $n$th-order jet space, denoted $g^{(n)}$ and $\mathbf{v}^{(n)}$, respectively. More details on prolonged group actions are discussed in Appendix A.2, with Figure 5 visualizing a simple example. To introduce our method, it suffices to note that the prolongation of the vector field (3) can be described explicitly by $\xi^j$ and $\phi$ and their derivatives via the *prolongation formula* (9).

## 3.2  Symbolic Regression Algorithms

Given the data $\{(x^i, y^i)\} \subset X \times Y$, the objective of symbolic regression (SR) is to find a symbolic expression for the function $y = f(x)$. Although this original formulation is for algebraic equations, it can be generalized to differential equations like (1). To discover a PDE from the dataset of its observed solutions on a grid $\Omega$, i.e. $\{(\mathbf{x}, u(\mathbf{x})) : \mathbf{x} \in \Omega\}$, we estimate the partial derivative terms and add them to the dataset: $\{(\mathbf{x}, u^{(n)}) : \mathbf{x} \in \Omega\}$. One of the variables in the variable set $(\mathbf{x}, u^{(n)})$ is used as the LHS of the equation, i.e. the role of the label $y$ in symbolic regression, while other variables serve as features. The precise set of derivatives added to symbolic regression and the choice of the equation LHS requires prior knowledge or speculations about the underlying system.

We briefly review two classes of SR algorithms: sparse regression (SINDy) and genetic programming (GP).

**Sparse regression** (Brunton et al., 2016) is specifically designed for discovering differential equations. It assumes the LHS $\ell$ of the equation is a fixed term, e.g. $\ell = u_t$, and the RHS of the equation can be written

as a *linear combination* of $m$ predefined functions $\theta^j$ with trainable coefficients $\mathbf{w} \in \mathbb{R}^m$, i.e.,

$$\ell(\mathbf{x}, u^{(n)}) = w^j \theta^j(\mathbf{x}, u^{(n)}), \ \theta^j : M^{(n)} \to \mathbb{R}. \tag{4}$$

The equation is found by solving for $\mathbf{w}$ that minimizes the objective $\|\mathbf{L} - \mathbf{R}\|_2^2 + \lambda\|\mathbf{w}\|_0$, where $\mathbf{L}$ and $\mathbf{R}$ are obtained by evaluating $\ell$ and $w^j \theta^j$ on all data points and concatenating them into column vectors, and $\|\mathbf{w}\|_0$ regularizes the number of nonzero terms. This formulation can be easily extended to $q$ equations and dependent variables ($q > 1$): $\ell^i(\mathbf{x}, \mathbf{u}^{(n)}) = W^{ij}\theta^j(\mathbf{x}, \mathbf{u}^{(n)})$, $W \in \mathbb{R}^{q \times m}$.

One problem with sparse regression is its restrictive assumptions about the form of equations. Many equations cannot be expressed in the form of (4), e.g. $y = \frac{1}{x+a}$ where $a$ could be any constant. Also, the success of sparse regression relies on the proper choice of the function library $\{\theta^j\}$. If any term in the true equation were not included, sparse regression would fail to identify the correct equation.

**Genetic programming** (GP) offers an alternative solution for SR (Cranmer, 2023), which can learn equations in more general forms. It represents each expression as a tree and instantiates a population of individual expressions. At each iteration, it samples a subset of expressions and selects one of them that best fits the data; the selected expression is then mutated by a random mutation, a crossover with another expression, or a constant optimization; the mutated expression replaces an expression in the population that does not fit the data well. The algorithm repeats this process to search for different combinations of variables, constants, and operators, and finally returns the "fittest" expression. GP can be less efficient than SINDy when the equation can be expressed in the form (4) due to its larger search space. However, we will show that it is a promising alternative to discover PDEs of generic forms, and our approach further boosts its efficiency.

## 4 Symbolic Regression with Symmetry Invariants

Symmetry offers a natural inductive bias for the search space of symbolic regression in differential equations. It reduces the dimensionality of the space and encourages parsimony of the resulting equations. To enforce symmetry in PDE discovery, we aim to find the maximal set of equations admitting a *given* symmetry and search in that set with symbolic regression (SR) methods.

### 4.1 Differential Invariants and Symmetry Conditions

To achieve this, our general strategy is to replace the original variable set with a complete set of *invariant functions* of the given symmetry group. Since we consider PDEs containing partial derivatives, the invariant functions refer to the *differential invariants* defined as follows.

**Definition 4.1** (Def. 2.51, Olver (1993)). Let $G$ be a local group of transformations acting on $X \times U$. Any $g \in G$ gives a prolonged group action $\mathrm{pr}^{(n)}g$ on the jet space $M^{(n)} \subset X \times U^{(n)}$. An $n$th order differential invariant of $G$ is a smooth function $\eta : M^{(n)} \to \mathbb{R}$, such that for all $g \in G$ and all $(\mathbf{x}, u^{(n)}) \in M^{(n)}$, $\eta(g^{(n)} \cdot (\mathbf{x}, u^{(n)})) = \eta(\mathbf{x}, u^{(n)})$ whenever $g^{(n)} \cdot (\mathbf{x}, u^{(n)})$ is defined.

In other words, differential invariants are functions of all variables and partial derivatives that remain invariant under prolonged group actions. Equivalently, if $G$ is generated by a set of infinitesimal generators $\mathcal{B} = \{\mathbf{v}_a\}$, then a function $\eta$ is a differential invariant of $G$ iff $\mathbf{v}_a^{(n)}(\eta) = 0$ for all $\mathbf{v}_a \in \mathcal{B}$. The following theorem guarantees that any differential equation admitting a symmetry group can be expressed solely in terms of the group invariants.

**Theorem 4.2** (Prop. 2.56, Olver (1993)). *Let $G$ be a local group of transformations acting on $X \times U$. Let $\{\eta^1(\mathbf{x}, u^{(n)}), ..., \eta^k(\mathbf{x}, u^{(n)})\}$ be a complete set of functionally independent $n$th-order differential invariants of $G$. An $n$th-order differential equation (1) admits $G$ as a symmetry group if and only if it is equivalent to an equation of the form $\tilde{F}(\eta^1, ..., \eta^k) = 0$.*

Consequently, SR with a complete set of invariants precisely searches within the space of all symmetric differential equations and automatically excludes equations violating the specified symmetry.

Our strategy of using differential invariants applies broadly to various equation discovery algorithms. For instance, in sparse regression, we can construct the function library using invariants rather than raw variables

and derivatives. Similarly, in genetic programming, the variable set can be redefined to include only invariant functions. In each case, the key benefit is the same: the search space is restricted to symmetry-respecting equations by construction. The reduced complexity of the equation search also leads to increased accuracy and efficiency.

Next, we describe how to construct a complete set of differential invariants (Section 4.2), and how to incorporate them into specific SR algorithms (Section 4.3).

## 4.2 Constructing a Complete Set of Invariants

Despite the simplicity of our strategy, we still need a concrete method for computing the invariants. In this subsection, we provide a general guideline to construct a complete set of differential invariants up to a required order given the group action.

By definition of differential invariants, we look for functions $\eta(\mathbf{x}, u^{(n)})$ satisfying $\mathbf{v}^{(n)}(\eta) = 0$ given a prolonged vector field $\mathbf{v}^{(n)}$. This is a first-order linear PDE that can be solved by the method of characteristics. However, in practice, if $E = X \times U \simeq \mathbb{R}^p \times \mathbb{R}$, there are $\binom{p+n-1}{n}$ partial derivatives of the independent variable $u$ of order exactly $n$. Therefore, as $n$ grows, it quickly becomes impractical to solve directly for $n$th-order differential invariants. The higher-order differential invariants, if necessary, can be computed recursively from lower-order ones by the following result:

**Proposition 4.3.** *Let $G$ be a local group of transformations acting on $X \times U \simeq \mathbb{R}^p \times \mathbb{R}$. Let $\eta^1, \eta^2, \cdots, \eta^p$ be any $p$ differential invariants of $G$ whose horizontal Jacobian $J = [D_i \eta^j]$ is non-degenerate on an open subset $\Omega \subset M^{(n)}$. If there are a maximal number of independent, strictly $n$th-order differential invariants $\zeta^1, \cdots, \zeta^{q_n}$, $q_n = \binom{p+n-1}{n}$, then the following set contains a complete set of independent, strictly $(n+1)$th-order differential invariants defined on $\Omega$:*

$$\det(D_i \tilde{\eta}^j_{(k,k')})/\det(D_i \eta^j), \ \forall k \in [p], k' \in [q_n], \tag{5}$$

*where $i, j \in [p]$ are matrix indices, $D_i$ denotes the total derivative w.r.t $i$-th independent variable and $\tilde{\eta}^j_{(k,k')} = [\eta^1, ..., \eta^{k-1}, \zeta^{k'}, \eta^{k+1}, ..., \eta^p]$.*

In practice, we first solve for pr $\mathbf{v}(\eta) = 0$ to obtain a sufficient number of lower-order invariants as required in Proposition 4.3, and then construct complete sets of invariants of arbitrary orders. Notably, while in theory our framework operates on any complete set of differential invariants, the invariants computed this way may be algebraically complicated and poorly scaled, leading to difficulties in SR optimization. In practice, we start from such a complete set of differential invariants and then deliberately convert them into simpler, physically interpretable invariant functions (such as Laplacians for rotational symmetry) as the feature set for SR. Then, we evaluate invariants on the dataset only where they are well-defined. If necessary, we shrink the domain and filter out data points that cause singularity (e.g., where the denominator of an invariant function vanishes). In Appendix A.4, we provide two examples of different symmetry groups and their differential invariants. Those results will also be used in our experiments.

## 4.3 Implementation in SR Algorithms

Our symmetry principle characterizes a subspace of all equations with a given symmetry. Generally, this subspace partially overlaps with the hypothesis spaces of SR algorithms, conceptually visualized in Figure 2. As in Theorem 4.2, PDEs with symmetry can be expressed as *implicit* functions of all differential invariants. However, symbolic regression methods typically learn *explicit* functions mapping features to labels. Some algorithms, such as SINDy, impose even stronger constraints on equation forms. Therefore, adaptation is needed to implement our strategy of using differential invariants in specific symbolic regression algorithms. In general, it requires more effort to adapt our method to base algorithms with more structured and nontrivial hypotheses about possible equation forms. Below, we discuss in detail how to adapt our method to two common classes of base SR algorithms.

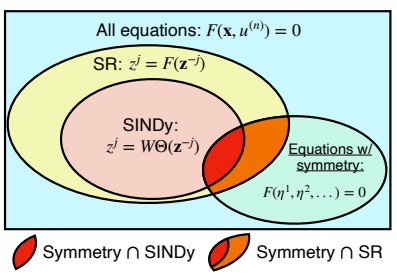

Figure 2: Venn diagram of hypothesis spaces from base SR methods and our symmetry principle.

**General explicit SR**  We start with general SR methods that learn an *explicit* function $y = f(x)$ without additional assumptions about the form of $f$, e.g., genetic programming and symbolic transformer. When learning the equation with differential invariants, we do not know which one of them should be used as the LHS of the equation, i.e. the label $y$ in symbolic regression. Thus, we fit an equation for each invariant as LHS and choose the equation with the lowest data error, as described in Algorithm 1. We use relative error to select the best equation since the scales of LHS terms differ.

---

**Algorithm 1** General explicit SR for differential equations with symmetry invariants

---

**Require:** PDE order $n$, dataset $\{\mathbf{z}^i = (\mathbf{x}^i, (u^{(n)})^i) \in M^{(n)}\}_{i=1}^{N_D}$, base SR algorithm $\mathcal{S} : (\mathbf{X}, \mathbf{y}) \mapsto y = f(x)$,
   infinitesimal generators of the symmetry group $\mathcal{B} = \{\mathbf{v}_a\}$.
**Ensure:** A PDE admitting the given symmetry group.
   Compute the symmetry invariants of $\mathcal{B}$ up to $n$th-order: $\eta^1, \cdots, \eta^K$. {Prop. 4.3}
   Evaluate the invariant functions on the dataset: $\eta^{k,i} = \eta^k(\mathbf{z}^i)$, for $k \in [K], i \in [N_D]$.
   Initialize a list of candidate equations and their risks: $\mathtt{E} = []$.
   **for** $k$ in $1:K$ **do**
      Use the $k$th invariant as label and the rest as features: $\mathbf{y} = \eta^{k,:}$, $\mathbf{X} = \eta^{-k,:}$.
      Run $\mathcal{S}(\mathbf{X}, \mathbf{y})$ and get a candidate equation $\eta^k = f^k(\boldsymbol{\eta}^{-k})$.
      Evaluate $\mathcal{L}^k = \|\mathbf{y} - f^k(\mathbf{X})\|_1 / \|\mathbf{y}\|_1$ and set $\mathtt{E}[k] = (f^k, \mathcal{L}^k)$.
   **end for**
   Choose the equation in $\mathtt{E}$ with the lowest error: $k = \arg\min_j \mathtt{E}[j][2]$.
   **return** $\eta^k = f^k(\boldsymbol{\eta}^{-k})$. {Optionally, expand all $\eta^j$ in terms of original variables $\mathbf{z}$.}

---

**Sparse regression**  SINDy assumes a linear equation form (4). Generally, its function library differs from the set of differential invariants. Also, SINDy fixes a LHS term, while we do not single out an invariant as the LHS of the equation when constructing the set of invariants.

Assume we are provided the SINDy configuration, i.e. the LHS term $\ell$ and the function library $\{\theta^j\}$. To implement sparse regression with symmetry invariants, we assign an invariant $\eta^k$ that symbolically depends on $\ell$, i.e. $\partial \eta^k / \partial \ell \neq 0$, as the LHS for the equation in terms of symmetry invariants. The remaining invariants are included on the RHS, where they serve as inputs of the original SINDy library functions. In other words, the equation form is $\eta^k = \tilde{w}^j \theta^j(\boldsymbol{\eta}^{-k})$. Similar to Algorithm 1, we can expand all $\eta$ variables to obtain the equation in original jet variables.

The above approach optimizes an unconstrained coefficient vector $\tilde{\mathbf{w}}$ for functions of symmetry invariants. Alternatively, we can use the original SINDy equation form (4) and implement the symmetry constraint as a constraint on the coefficient $\mathbf{w}$, as demonstrated in the following theorem. Here, we generalize the setup to multiple dependent variables and equations.

**Proposition 4.4.** *Let $\boldsymbol{\ell}(\mathbf{x}, \mathbf{u}^{(n)}) = W\boldsymbol{\theta}(\mathbf{x}, \mathbf{u}^{(n)})$ be a system of $q$ differential equations admitting a symmetry group $G$, where $\mathbf{x} \in \mathbb{R}^p$, $\mathbf{u} \in \mathbb{R}^q$, $\boldsymbol{\theta} \in \mathbb{R}^m$. Assume there exist some $n$th-order invariants of $G$, $\eta_0^{1:q}$ and $\eta^{1:K}$, s.t. (1) the system of equations can be expressed as $\boldsymbol{\eta}_0 = W'\boldsymbol{\theta}'(\boldsymbol{\eta})$, where $\boldsymbol{\eta}_0 = [\eta_0^{1:q}]$ and $\boldsymbol{\eta} = [\eta^{1:K}]$, and (2) $\eta_0^i = T^{ijk}\theta^k \ell^j$ and $(\theta')^i = S^{ij}\theta^j$, for some functions $\boldsymbol{\theta}'(\boldsymbol{\eta})$ and constant tensors $W'$, $T$ and $S$. Then, the space of all possible $W$ is a linear subspace of $\mathbb{R}^{q \times m}$.*

Intuitively, the conditions above state that the equations can be expressed as a linear combination of invariant terms, similar to the form in (4) w.r.t original jet variables. Also, every invariant term in $\boldsymbol{\eta}_0$ and $\boldsymbol{\theta}'(\boldsymbol{\eta})$ is already encoded in the original library $\boldsymbol{\theta}$. In practice, we need to choose a suitable set of invariants according to the SINDy configuration to meet these conditions. For example, it is a common SINDy setup where $\boldsymbol{\theta}$ contains all monomials on $M^{(n)}$ up to degree $d$. In this case, any set of invariants where each invariant is a polynomial on $M^{(n)}$ up to degree $d$ satisfies these conditions.

The proof of Proposition 4.4 is provided in Appendix A.5, where we explicitly identify the basis of the linear subspace for $W$. Then, we can use this basis to build a SINDy model for PDEs with the corresponding symmetry, similar to how EMLP (Finzi et al., 2021b) constructs equivariant linear layers and how Yang et al. (2024) constructs equivariant SINDy for ODEs. Specifically, if the constrained subspace has a basis

$Q \in \mathbb{R}^{r \times q \times m}$, where $r$ is the subspace dimension, we write $W^{jk} = Q^{ijk}\beta^i$. We then fix $Q$ and solve $\beta \in \mathbb{R}^r$ using the same least square optimizer in SINDy, effectively reducing the parameter space dimension from $q \times m$ to $r$.

In practice, we observe that the basis $Q$ obtained from the constructive proof of Proposition 4.4 is not sparse. The lack of sparsity can pose a problem when we perform sequential thresholding in sparse regression. Specifically, in SINDy, the entries in $W$ that are close to zero are filtered out at the end of each iteration, which serves as a proxy to the sparsity-promoting $L_0$ regularization. Since we fix $Q$ and only optimize $\beta$, a straightforward modification to the sequential thresholding procedure is to threshold the entries in $\beta$ instead of those in $W$. However, if $Q$ is dense, even a sparse vector $\beta$ can lead to a dense $W$, which contradicts the purpose of sparse regression. To address this issue, we apply a Sparse PCA to $Q$ to obtain a sparsified basis. More implementation details and examples of the computed basis $Q$ can be found in Appendix B.2.

One notable advantage of converting symmetry into linear constraints with Proposition 4.4 is that it allows us to keep track of the original SINDy parameters $W$ during optimization. This enables straightforward integration of symmetry constraints to variants of SINDy, e.g. Weak SINDy (Messenger & Bortz, 2021a;b) for noisy data. For $W^{jk} = Q^{ijk}\beta^i$, while we directly optimize $\beta$, we can still easily compute the objective of Weak SINDy which explicitly depends on $W$. In comparison, if we use the raw invariant terms for regression, e.g. the equations take the form $\boldsymbol{\eta}_0 = W'\boldsymbol{\theta}'(\boldsymbol{\eta})$, it is challenging to formulate the objective of Weak SINDy with respect to $W'$. In Appendix B.3, we provide more details on how to implement this linear-constraint approach on Weak SINDy.

### 4.4 Constraint Relaxation for Systems with Imperfect Symmetry

Our approach discovers PDEs assuming perfect symmetry. However, it is common in reality that a system shows imperfect symmetry due to external forces, boundary conditions, etc. (Wang et al., 2022). In such cases, the previous method cannot identify any symmetry-breaking factors.

To address this, we propose to relax the symmetry constraints by allowing symmetry-breaking terms to appear in the equation, but at a higher "cost". We implement this idea in sparse regression, where the equation has a linear structure $\boldsymbol{\ell} = W\boldsymbol{\theta}$. We adopt the technique from Residual Pathway Prior (RPP) (Finzi et al., 2021a), which is originally developed for equivariant linear layers in neural networks. Specifically, let $Q$ be the basis of the parameter subspace that preserves symmetry and $P$ be the orthogonal complement of $Q$. Instead of parameterizing $W$ in this subspace, we define $W = A + B$ where $A^{jk} = Q^{ijk}\beta^i$ and $B^{jk} = P^{ijk}\gamma^i$, where $\beta$ and $\gamma$ are learnable parameters, and place a stronger regularization on $\gamma$ than on $\beta$. While the model still favors equations in the symmetry subspace spanned by $Q$, symmetry-breaking components in $P$ can appear if it fits the data well.

## 5 Experiments

### 5.1 Datasets and Their Symmetries

We consider the following PDE systems, which cover different challenges in PDE discovery, such as high-order derivatives, generic equation form, multiple dependent variables and equations, noisy dataset, and imperfect symmetry. The datasets are generated by simulating the ground truth equation from specified initial conditions, with detailed procedures described in Appendix E.1.

**Boussinesq Equation.** Consider the Boussinesq equation describing the unidirectional propagation of a solitary wave in shallow water (Newell, 1985):

$$u_{tt} + uu_{xx} + u_x^2 + u_{xxxx} = 0 \tag{6}$$

This equation has a scaling symmetry $\mathbf{v}_1 = 2t\partial_t + x\partial_x - 2u\partial_u$ and the translation symmetries in space and time. The differential invariants are given by $\eta_{(\alpha,\beta)} = u_{x^{(\alpha)}t^{(\beta)}} u_x^{-(2+\alpha+2\beta)/3}$ where $\alpha$ and $\beta$ are the orders of partial derivatives in $x$ and $t$, respectively. To discover the 4th-order equation, we compute all $\eta_{(\alpha,\beta)}$ for $0 \leq \alpha + \beta \leq 4$, except for $\eta_{(1,0)} = 1$ which is a constant.

**Darcy Flow.** The following PDE describes the steady state of a 2D Darcy flow (Takamoto et al., 2022) with spatially varying viscosity $a(x, y) = e^{-4(x^2+y^2)}$ and a constant force term $f(x) = 1$:

$$-\nabla(e^{-4(x^2+y^2)}\nabla u) = 1 \tag{7}$$

This equation admits an SO(2) rotation symmetry $\mathbf{v} = y\partial_x - x\partial_y$. A detailed calculation of the differential invariants of this group can be found in Example A.5. In our experiment, we use the following complete set of 2nd-order invariants: $\{\frac{1}{2}(x^2+y^2), u, xu_y - yu_x, xu_x + yu_y, u_{xx} + u_{yy}, u_{xx}^2 + 2u_{xy}^2 + u_{yy}^2, x^2u_{xx} + y^2u_{yy} + 2xyu_{xy}\}$.

**Reaction-Diffusion.** We consider the following system of PDEs from Champion et al. (2019):

$$
\begin{aligned}
u_t &= d_1\nabla^2 u + (1 - u^2 - v^2)u + (u^2 + v^2)v \\
v_t &= d_2\nabla^2 v - (u^2 + v^2)u + (1 - u^2 - v^2)v
\end{aligned}
\tag{8}
$$

In the default setup, we use $d_1 = d_2 = 0.1$. The system then exhibits rotational symmetry in the phase space: $\mathbf{v} = u\partial_v - v\partial_u$. The ordinary invariants (functions of variables, not derivatives) are $\{t, x, y, u^2 + v^2\}$. The higher-order invariants are $\{\mathbf{u} \cdot \mathbf{u}_\mu, \mathbf{u}^\perp \cdot \mathbf{u}_\mu\}$, where $\mathbf{u} = (u, v)^T$ and $\mu$ is any multi-index of $t$, $x$ and $y$.

We also consider the following cases where the rotation symmetry is broken due to different factors:

- **Unequal diffusivities** We use different diffusion coefficients for the two components: $d_1 = 0.1$, $d_2 = 0.1 + \epsilon$. This can happen, for example, when two chemical species described by the equation diffuse at different rates due to molecular size, charge, or solvent interactions.

- **External forcing** The ground truth equation (8) is modified by adding $-\epsilon v$ to the RHS of $u_t$ and $-\epsilon u$ to the RHS of $v_t$. This can reflect a weak parametric forcing on the system.

## 5.2 Methods and Evaluation Criteria

We consider three classes of algorithms for equation discovery: sparse regression (PySINDy, de Silva et al. (2020); Kaptanoglu et al. (2022)), genetic programming (PySR, Cranmer (2023)), and a pretrained symbolic transformer (E2E, Kamienny et al. (2022)). For each class, we compare the original algorithm using the regular jet space variables (i.e. $(\mathbf{x}, u^{(n)})$ ) and our method using symmetry invariants. Our method will be referenced as SI (**S**ymmetry **I**nvariants) in the results.

To evaluate an equation discovery algorithm, we run it 100 times with randomly sampled data subsets and randomly initialized models if applicable. We record its *success probability* (SP) of discovering the correct equation. Specifically, we expand the ground truth equation into $\sum_i c^i f^i(\mathbf{z}) = 0$, where $c^i$ are nonzero coefficients, $\mathbf{z}$ denotes the variables involved in the algorithm, i.e., original jet variables $(\mathbf{x}, u^{(n)})$ for baselines and symmetry invariants for our method, and $f^i$ are functions of $\mathbf{z}$. Also, the discovered equation is expanded as $\sum_i \hat{c}^i \hat{f}^i(\mathbf{z}) = 0$, where $\hat{c}^i \neq 0$. The discovered equation is considered correct if all the terms with nonzero coefficients match the ground truth, i.e., $\{f^i\} = \{\hat{f}^i\}$. We also report the *prediction error* (PE), which measures how well the discovered equation fits the data. For evolution equations with time derivatives on the LHS, we simulate each discovered equation from an initial condition and measure its difference from the ground truth solution at a specific timestep in terms of root mean square error (RMSE). Otherwise, we just report the RMSE of the discovered equation evaluated on all test data points.

### 5.3 Results on Clean Data with Perfect Symmetry

Table 1: Equation discovery results on clean data. $\mathcal{C}$, standing for *complexity*, refers to the effective parameter space dimension in sparse regression and the number of variables in GP/Transformer. SP and PE stands for *success probability* and *prediction error*, as explained in Section 5.2. The entries "-" suggest that the method does not apply to the specific PDE system, or the result is not meaningful. The arrows $\uparrow$ / $\downarrow$ mean higher/lower metrics are better. Confidence intervals for PE are reported in Table 3.

| Method | | Boussinesq (6) | | | Darcy flow (7) | | | Reaction-diffusion (8) | | |
| | | $\mathcal{C} \downarrow$ | SP $\uparrow$ | PE $\downarrow$ | $\mathcal{C} \downarrow$ | SP $\uparrow$ | PE $\downarrow$ | $\mathcal{C} \downarrow$ | SP $\uparrow$ | PE $\downarrow$ |
|---|---|---|---|---|---|---|---|---|---|---|
| Sparse | PySINDy | 15 | 0.00 | 0.373 | - | - | - | 38 | 0.53 | 0.021 |
| Regression | SI | 15 | **1.00** | **0.098** | - | - | - | 28 | **0.54** | **0.008** |
| Genetic | PySR | 17 | 0.90 | 0.098 | 8 | 0.00 | 0.114 | 17 | 0.00 | - |
| Programming | SI | **14** | **1.00** | 0.098 | **7** | **0.79** | **0.051** | **16** | **0.81** | **0.023** |
| Transformer | E2E | 10 | 0.53 | 0.132 | 8 | 0.00 | - | 17 | 0.00 | - |
| | SI | **7** | **0.85** | **0.104** | **7** | 0.00 | - | **16** | 0.00 | - |

Table 1 summarizes the performance of all methods on the three PDE systems. For prediction errors (PE), we report the median, instead of the mean + standard deviation, of 100 runs for each algorithm. This is because some incorrectly discovered equations yield large prediction errors, making the mean and the standard deviation less meaningful. We additionally report the confidence intervals for prediction errors by [25% quantile, 75% quantile] in Table 3. Comparisons are made within each class of methods. Generally, using symmetry invariants reduces the complexity, defined as the effective parameter space dimension in sparse regression and the number of variables in GP/Transformer, of equation discovery, and improves the chance of finding the correct equations compared to the baselines.

Specifically, in sparse regression, our method (SI) using symmetry invariants is only slightly better than PySINDy in the reaction-diffusion system, but constantly succeeds in the Boussinesq equation where PySINDy fails. The failure of PySINDy is because the $u_x^2$ term in (6) is not supported by its function library, showing that SINDy's success relies heavily on the choice of function library. This exclusion of function terms also explains why the reported complexity ($\mathcal{C}$) of PySINDy is the same as that of our method (SI), which would otherwise be smaller because symmetry reduces the hypothesis space of equations. On the other hand, by enforcing the equation to be expressed in invariants, our method automatically identifies the proper function library. Appendix C.2 provides results for other variants of sparse regression, where we modify the implementation of PySINDy to include different terms in its library.

For GP-based methods, Table 1 displays the results with a fixed number of GP iterations for each dataset. We also experiment with different numbers of iterations in Figure 3, showing that our method can identify the correct equations with fewer iterations and is considered more efficient.

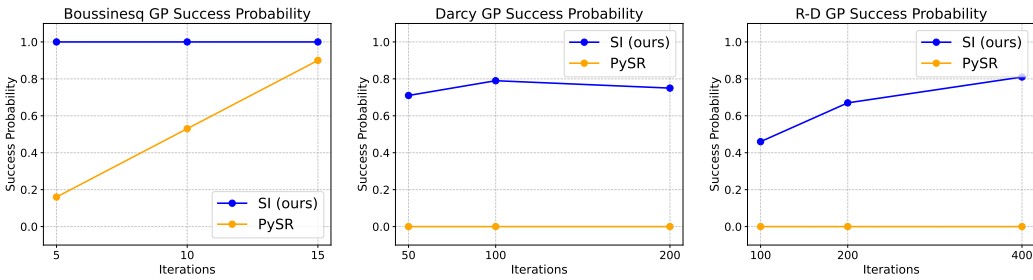

Figure 3: Success Probabilities of GP-based methods on different systems. Our method with symmetry invariants can discover the correct equations with fewer iterations.

On the other hand, the pretrained symbolic transformer fails on two of the three datasets. We conjecture this is because the data distribution from PDE solutions greatly differs from its pretraining dataset. However, the symbolic transformer can discover the Boussinesq equation correctly, where using symmetry invariants leads to a much higher success probability.

### 5.4 Results on Noisy Data and Imperfect Symmetry

We test the robustness of our method under two challenging scenarios: (1) noise in observed data, and (2) PDE with imperfect symmetry.

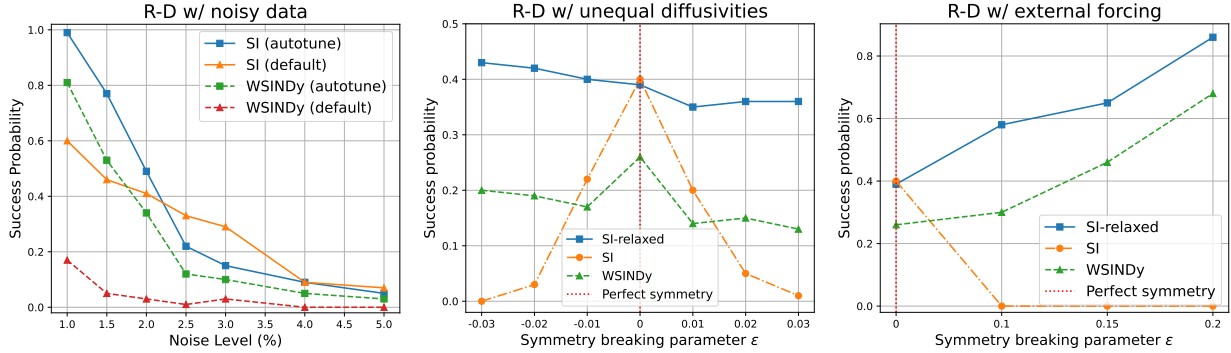

Figure 4: Success probabilities of sparse regression methods on the reaction-diffusion system with noisy data (left), unequal diffusivities (center) and external forcing (right). Under noisy data, our method (SI) consistently outperforms WSINDy under the same test function configuration. For systems with imperfect symmetry, strictly enforcing symmetry (SI) hurts performance, but a relaxed symmetry constraint (SI-relaxed, introduced in Section 4.4) is still better than no inductive bias (WSINDy).

In the first experiment, we add different levels of white noise to the simulated solution of the reaction-diffusion system. Since the derivatives estimated by finite difference are inaccurate with the noisy solution, we use WSINDy (Messenger & Bortz, 2021a), which does not require derivative estimation. The success probabilities of our method (SI) and WSINDy are shown in Figure 4 (left). We adopt two different strategies for choosing the test function parameters in WSINDy, including the number of test functions, half-lengths of the square subdomains in each spatiotemporal direction, and the polynomial degree of test functions: (1) following Messenger & Bortz (2021a), the parameters are *auto-tuned* based on the noise characteristic of the data; and (2) the *default* parameters in PySINDy are used. In both WSINDy setups, our method consistently achieves a higher success probability at different noise levels. Notably, when the noise level is high, our symmetry-constrained model performs better with the default parameter setup, which, as we observe, uses fewer test functions and lower polynomial degrees than the auto-tuned setup. We comment that choosing test functions and related hyperparameters is known to be a challenging problem (Bortz et al., 2023; Tran & Bortz, 2025), and we leave further investigation of this phenomenon to future work.

In the second experiment, we simulate the two variants of (8) (unequal diffusivities and external forcing) with different values for the symmetry-breaking parameter $\epsilon$ and add 2% noise to the numerical solutions. We compare three models: (1) our model with strictly enforced symmetry (SI), (2) our model with relaxed symmetry (SI-relaxed) introduced in Section 4.4, and (3) WSINDy with default PySINDy parameters as the baseline. The results for the two systems with symmetry breaking are shown in Figure 4 (center & right). As expected, SI has a much lower success probability when the symmetry-breaking factor becomes significant. Meanwhile, SI-relaxed remains highly competitive. It also has a clear advantage over baseline SINDy, showing that even if the inductive bias of symmetry is slightly inaccurate, our model with relaxed constraints is still better than a model without any knowledge of symmetry. We also include the results on the same systems with auto-tuned WSINDy parameters in Appendix C.3.

More comprehensive results, e.g. variant sparse regression models, comparison with an additional D-CIPHER (Kacprzyk et al., 2023) baseline, equations with higher spatial dimensions and larger symmetry groups, discovered equation samples, are provided in Appendix C.

## 6 Discussion

We propose to enforce symmetry in symbolic regression algorithms for discovering PDEs by using differential invariants of the symmetry group as the variable set. We implement this general strategy in different classes of algorithms and observe improved accuracy, efficiency and robustness of equation discovery, especially in challenging scenarios such as noisy data and imperfect symmetry.

It should be noted that our method assumes the symmetry group is already given. This assumption aligns with common practice: physicists often begin by hypothesizing the symmetries of a system and seek governing equations allowed by those symmetries. However, our current framework cannot be applied if symmetry is unknown, and will produce incorrect results with misspecified symmetry. This can be potentially addressed by incorporating automated symmetry discovery methods for differential equations (Yang et al., 2024; Ko et al., 2024), which we leave for future work.

Another caveat of our method is the calculation of differential invariants. While solving for $\mathbf{v}^{(n)}(\eta) = 0$ and applying the formula (5) is easy with any symbolic computation package, the resulting differential invariants may be complicated and require ad-hoc adjustment for better interpretability and compatibility with specific algorithm implementations (e.g. conditions in Proposition 4.4). Fortunately, this only requires a one-time effort. Once we have derived the invariants for a symmetry group, the results can be reused for any equation admitting the same symmetry.

### Broader Impact Statement

The method in this paper can potentially be used to expedite the process of discovering governing equations from data and aid researchers in other scientific domains. Equally important, equations inferred from imperfect or biased data may appear authoritative yet embed systematic errors. Thorough validation checks, uncertainty quantification, and domain-expert review protocols for the discovered equations are essential.

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

# A  Math

## A.1  Notations

Table 2: Descriptions of symbols used throughout the paper. The three blocks include (1) basic notations for PDEs, (2) notations for Lie symmetry of PDEs, and (3) notations for symbolic regression algorithms and miscellaneous.

| Symbols | Descriptions |
|---|---|
| $p$ | Number of independent variables of a PDE. |
| $q$ | Number of dependent variables of a PDE. |
| $X$ | Space of independent variables of a PDE: $X \subset \mathbb{R}^p$. Also used to denote the feature space of SR algorithms. |
| $U$ | Space of dependent variables of a PDE: $U \subset \mathbb{R}^q$. Assumed to be 1-dimensional unless otherwise stated. |
| $E$ | Total space of all variables of a PDE: $E = X \times U$. |
| $U_k$ | Space of strictly $k$th-order partial derivatives of variables in $U$ w.r.t variables in $X$. |
| $U^{(n)}$ | Space of all partial derivatives up to $n$th order (including the original variables in $U$): $U^{(n)} = U \times U_1 \times \cdots \times U_n$. |
| $M^{(n)}$ | $n$th-order jet space: $M^{(n)} \subset X \times U^{(n)}$. |
| $T\mathcal{M}$ | The tangent bundle of a manifold $\mathcal{M}$. |
| $\mathbf{x}$ | Independent variables of a PDE: $\mathbf{x} \in \mathbb{R}^p$. |
| $t$ | Time variable. |
| $x, y$ | Spatial variables in PDE contexts. Also used to denote the features and labels of SR algorithms, where $x$ can denote multi-dimensional features. |
| $u, \mathbf{u}$ | Dependent variable(s) of a PDE: $u \in \mathbb{R}$ and $\mathbf{u} \in \mathbb{R}^q$. |
| $u^{(n)}, \mathbf{u}^{(n)}$ | The collection of all up to $n$-th order partial derivatives of $u$ or $\mathbf{u}$. |
| $df$ | The (ordinary) differential of a function. For a differential function $f : M^{(n)} \to \mathbb{R}$, $df = \sum_j \frac{\partial f}{\partial x^j} dx^j + \sum_\alpha \frac{\partial f}{\partial u_\alpha} du_\alpha$. |
| $D_i f$ | The total derivative of a differential function $f : M^{(n)} \to \mathbb{R}$ w.r.t the $i$th independent variable. For example, if $p = q = 1$, $D_1 f = \frac{\partial f}{\partial x} + \sum_{k=0}^{\infty} u_{k+1} \frac{\partial f}{\partial u_k}$, where $u_k := \partial^k u / \partial x^k$. |
| $Df$ | The total differential of a differential function $f : M^{(n)} \to \mathbb{R}$, i.e. $Df = D_i f \ dx^i$. |
| $g$ | A group element with an action on $E$ (2). |
| $\mathbf{v}$ | A vector field on the total space $E$ (3), representing an infinitesimal transformation. A list of multiple vector fields are indexed by subscripts. |
| $\mathrm{pr}^{(n)} g$ | $n$th-order prolongation of $g$ acting on $M^{(n)}$. |
| $\mathrm{pr}^{(n)} \mathbf{v}$ | $n$th-order prolongation of $\mathbf{v}$ acting on $M^{(n)}$. |
| $g^{(n)}, \mathbf{v}^{(n)}$ | Equivalent to $\mathrm{pr}^{(n)} g$ and $\mathrm{pr}^{(n)} \mathbf{v}$, respectively. |
| $\mathrm{pr}\ \mathbf{v}$ | The (infinite) prolongation of $\mathbf{v}$. For an $n$th-order differential function $f(\mathbf{x}, \mathbf{u}^{(n)})$, $\mathrm{pr}\ \mathbf{v}(f) = \mathrm{pr}^{(n)} \mathbf{v}(f)$. |
| $\eta, \zeta, \vartheta$ | Differential invariants of a symmetry group. $\eta$ is used by default. The other letters are used to distinguish between invariants of different orders. |
| $\ell, \boldsymbol{\ell}$ | The LHS of SINDy equation (4). Often assumed to be time derivatives. |
| $\boldsymbol{\theta}$ | A column vector containing all SINDy library functions: $\boldsymbol{\theta} = [\theta^1, \cdots, \theta^m]$ |
| $\mathbf{w}, W$ | The SINDy parameters. For only one equation, $\mathbf{w} = [w^1, \cdots, w^m]$ is a row vector. For multiple equations, $W = [w^{ij}]$ is a $q \times m$ matrix. |
| $\mathbf{X}, \mathbf{y}$ | Concatenated matrix/vector of features/labels of all datapoints for symbolic regression. |
| $[N]$ | List of positive integers up to $N$, i.e. $[1, 2, \cdots, N]$ for any $N \in \mathbb{Z}^+$. |
| $1 : N$ | Equivalent to $[N]$. |
| LHS, RHS | Left- and Right-hand side of an equation. |

## A.2 Extended Background on PDE Symmetry

References for the below material include Olver (1993), Olver (1995).

**Prolonged group actions**  Let $E = X \times U \simeq \mathbb{R}^p \times \mathbb{R}^q$ be endowed with the action of a group $G$ via point transformations. Then group elements $g \in G$ act locally on functions $\mathbf{u} = f(\mathbf{x})$, therefore also on derivatives of these functions. This in turn induces, at least pointwise, "prolonged" transformations on jet spaces: $(\tilde{\mathbf{x}}, \tilde{\mathbf{u}}^{(n)}) = \mathrm{pr}^{(n)} g \cdot (\mathbf{x}, \mathbf{u}^{(n)})$.

Let $J = (j_1, \ldots, j_n), 1 \leq j_\nu \leq p$ be an $n$-tuple of indices of independent variables and $1 \leq \alpha \leq q$. We will use the shorthand

$$u_J^\alpha := \frac{\partial^J u^\alpha}{\partial x^J} := \frac{\partial^{|J|} u^\alpha}{\partial x_{j_1} \cdots \partial x_{j_n}}$$

and

$$D_J := D_{j_1} \cdots D_{j_n}.$$

It is not practical to work explicitly with prolonged group transformations. Therefore one linearizes and considers the prolonged action of the infinitesimal generators of $G$. Explicitly, given a vector field

$$\mathbf{v} = \sum_{i=1}^p \xi^i(\mathbf{x}, \mathbf{u}) \frac{\partial}{\partial x^i} + \sum_{\alpha=1}^q \varphi^\alpha(\mathbf{x}, \mathbf{u}) \frac{\partial}{\partial u^\alpha},$$

its **characteristic** is a $q$-tuple $Q = (Q^1, \ldots, Q^q)$ of functions with

$$Q^\alpha(\mathbf{x}, \mathbf{u}^{(1)}) = \varphi^\alpha(\mathbf{x}, \mathbf{u}) - \sum_{i=1}^p \xi^i(\mathbf{x}, \mathbf{u}) \frac{\partial u^\alpha}{\partial x^i}.$$

Now the prolongation of $\mathbf{v}$ to order $n$ is defined by

$$\mathrm{pr}^{(n)} \mathbf{v} = \sum_{i=1}^p \xi^i(\mathbf{x}, \mathbf{u}) \frac{\partial}{\partial x^i} + \sum_{\alpha=1}^q \sum_{\#J=n} \varphi_J^\alpha(\mathbf{x}, \mathbf{u}^{(n)}) \frac{\partial}{\partial u_J^\alpha}. \tag{9}$$

Here $J$ ranges over all $n$-tuples $J = (j_1, \ldots, j_n), 1 \leq j_\nu \leq p$ and the $\varphi_J^\alpha$ are given by

$$\varphi_J^\alpha = D_J Q^\alpha + \sum_{i=1}^p \xi^i \mathbf{u}_{J,i}^\alpha.$$

We remark that the prolongation of $\mathbf{v}$ has been described explicitly in terms of the coefficients of $\mathbf{v}$ and their derivatives.

Figure 5 visualizes the group action of a Lie point transformation and its prolongation with a simple example. Consider the total space $X \times U \simeq \mathbb{R} \times \mathbb{R}$, and the standard rotation generator in 2D space given by $\mathbf{v} = -u\partial_x + x\partial_u$. The vector field is visualized in dark red arrows in the background. We also consider a function $X \to U$ given by $u(x) = 0.5(x-1)^3$, whose graph is visualized by the dark red solid line in Figure 5 left. The graph of its first-order derivative, $u_x(x) = 1.5(x-1)^2$, is visualized by the dark green dash-dot line.

Then, we choose a random group element $g = \exp(\theta \mathbf{v})$ that rotates a 2D vector $(x, u) \in X \times U$ by angle $\theta$. Applying this pointwise transformation to every point on the graph of $u(x)$, we have a transformed graph visualized by the dark red dashed line. The transformed function, $\tilde{u} = g \cdot u$, is defined as the function whose graph is the transformed graph. In other words, $\tilde{u}(x) = (g \cdot u)(x)$ is visualized by the dark red dashed line in Figure 5 left.

Next, we consider how the rotation of $(x, u)$ transforms the first-order derivative $u_x := \frac{\mathrm{d}u}{\mathrm{d}x}$. The prolonged vector field, i.e., the infinitesimal generator of the prolonged group action, can be computed by (9): $\mathbf{v}^{(1)} = \mathbf{v} + (1 + u_x^2)\partial_{u_x}$. The projection of $\mathbf{v}^{(1)}$ onto $X \times U_1$ is visualized in the dark green arrows in Figure 5

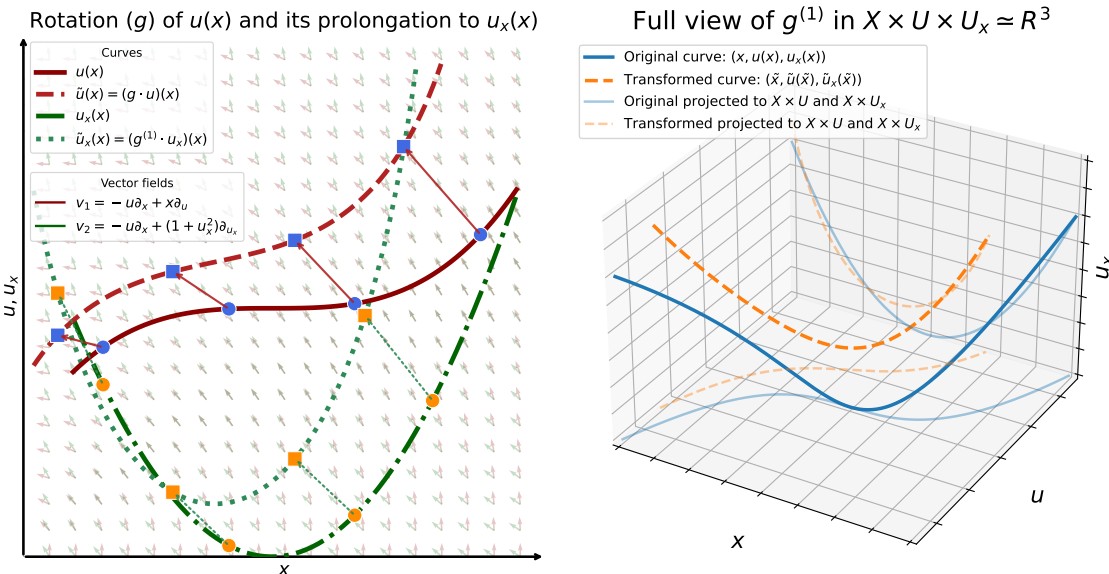

Figure 5: Demonstration of the rotation $\mathbf{v} = -u\partial_x + x\partial_u$ acting on $X \times U$, and its first-order prolongation acting on $X \times U \times U_1$.

left. Similarly, the prolonged group action $g^{(1)} = \exp(\theta\mathbf{v}^{(1)})$ is applied to every point on the graph of $u_x(x)$, yielding the graph of the transformed derivative function, $\tilde{u}_x(x) = \frac{\mathrm{d}\tilde{u}}{\mathrm{d}x}(x)$, visualized in the dark green dotted line.

The full transformation of the prolonged $g^{(1)}$ in the 3D space $X \times U \times U_1$ is shown on Figure 5 right. The graph of the original prolonged function $u^{(1)}(x) = (u(x), u_x(x))$ is shown in the solid line, which is transformed into the dashed line by $g^{(1)}$.

### A.3 Proof of Proposition 4.3

Olver (1995) provides the following general theorem to construct higher-order differential invariants from a contact-invariant coframe. We refer the readers to Chapter 5 of Olver (1995) for definitions of relevant concepts, e.g., contact forms and contact-invariant forms and coframes.

**Theorem A.1** (Thm. 5.48, (Olver, 1995)). *Let $G$ be a transformation group acting on a space with $p$ independent variables and $q$ dependent variables. Suppose $\omega^1, ..., \omega^p$ is a contact-invariant coframe for $G$, and let $\mathcal{D}_j$ be the associated invariant differential operators defined via $Df = D_j f \; dx^j = \mathcal{D}_j f \; \omega^j$. If there are a e number of independent, strictly $n$th-order differential invariants $\zeta^1, \cdots, \zeta^{q_n}$, $q_n = \binom{p+n-1}{n}$, then the set of differentiated invariants $\mathcal{D}_i\zeta^\nu$, $i \in [p]$, $\nu \in [q_n]$, contains a complete set of independent, strictly $(n+1)$th-order differential invariants.*

Specifically, the condition that there exist a maximal number of differential invariants of order exactly $n$ is guaranteed if $n$ is at least $\dim G$.

Our proposition is a derived result from the above theorem, which provides a concrete way of computation from lower-order invariants to higher-order ones:

**Proposition A.2.** *Let $G$ be a local group acting on $X \times U \simeq \mathbb{R}^p \times \mathbb{R}$. Let $\eta^1, \eta^2, \cdots, \eta^p$ be any $p$ differential invariants of $G$ whose horizontal Jacobian $J = [D_i\eta^j]$ is non-degenerate on an open subset $\Omega \subset M^{(n)}$. If there are a maximal number of independent, strictly $n$th-order differential invariants $\zeta^1, \cdots, \zeta^{q_n}$, $q_n = \binom{p+n-1}{n}$, then the following set contains a complete set of independent, strictly $(n+1)$th-order differential invariants defined on $\Omega$:*

$$\frac{\det(D_i\tilde{\eta}^j_{(k,k')})}{\det(D_i\eta^j)}, \; \forall k \in [p], k' \in [q_n], \tag{10}$$

where $i, j \in [p]$ are matrix indices, $D_i$ denotes the total derivative w.r.t $i$-th independent variable and $\tilde{\eta}^j_{(k,k')} = [\eta^1, ..., \eta^{k-1}, \zeta^{k'}, \eta^{k+1}, ..., \eta^p]$.

*Proof.* We show that the total differentials of the differential invariants $\eta^1, ..., \eta^p$ can be used to construct a *contact-invariant coframe* of $G$ and then derive the associated invariant differential operators to complete the proof.

First, note that for any differential invariant $\eta$ of $G$, its total differential $\omega = D\eta = D_j\eta \; dx^j$ can be written as

$$\omega = \omega_o + \theta, \tag{11}$$

where $\omega_o := d\eta = \sum_{i \in [p]} \frac{\partial F}{\partial x^i} dx^i + \sum_{|\alpha| \le n} \frac{\partial F}{\partial u_\alpha} du_\alpha$ is the ordinary differential of $\eta : M^{(n)} \to \mathbb{R}$ and $\theta$ is a contact form.

Since $\eta$ is a differential invariant, its differential $\omega_o = d\eta$ is an invariant one-form on $M^{(n)}$, i.e. $(g^{(n)})^*\omega_o = \omega_o$.

Also, a prolonged group action maps contact forms to contact forms. To see this, note that a prolonged group action $g^{(n)}$ maps the prolonged graph of any function to the prolonged graph of a transformed function. Then, for any contact form $\theta$, $(g^{(n)})^*\theta$ is annihilated by all prolonged functions $f^{(n)}$, thus a contact form by definition:

$$\begin{aligned}
(f^{(n)})^*((g^{(n)})^*\theta) &= (g^{(n)} \circ f^{(n)})^*\theta \\
&= ((g \cdot f)^{(n)})^*\theta \\
&= 0.
\end{aligned} \tag{12}$$

Then, from (11), we have

$$\begin{aligned}
(g^{(n+1)})^*\omega &= (g^{(n)})^*\omega_o + (g^{(n+1)})^*\theta \\
&= \omega_o + \theta' \\
&= \omega + (\theta' - \theta)
\end{aligned} \tag{13}$$

where $\theta'$ is some contact form and so is $\theta' - \theta$. Thus, $\omega$ is contact-invariant. For the $p$ differential invariants $\eta^1, \cdots, \eta^p$, we have $p$ contact-invariant one-forms $\omega^1, \cdots, \omega^p$, respectively.

Next, we prove that $\omega^1, \cdots, \omega^p$ are linearly independent and form a coframe. Assume there exists smooth coefficients $c^j$ such that $\sum_j c^j \omega^j = 0$. Then, regrouping the coefficients of the horizontal forms $dx^i$, we have

$$0 = \sum_{i,j} c^j D_i \eta^j dx^i = \sum_i \left( \sum_j c^j D_i \eta^j \right) dx^i. \tag{14}$$

Because the $dx^i$ are linearly independent, each coefficient of $dx^i$ must vanish, i.e. $J_i{}^j c^j = 0$. Since the Jacobian $J = [D_i \eta^j]$ is non-degenerate, the only solution is $c^j = 0$ (on the open subset $\Omega \in M^{(n)}$). Thus, $\omega^1, \cdots, \omega^p$ form a contact-invariant coframe. According to Theorem A.1, the associated invariant differential operators of the coframe take a complete set of same-order invariants to a complete set of one-order-higher invariants.

The remaining step is to obtain the invariant differential operators explicitly in terms of $\eta^j$. Recall the formula in Theorem A.1 that defines the invariant differential operators:

$$D_i f \; dx^i = \mathcal{D}_j f \; \omega^j. \tag{15}$$

Expanding $\omega^j = D\eta^j = D_i \eta^j \; dx^i$, we have the following linear system of invariant differential operators $\mathcal{D}_j$:

$$\begin{bmatrix} D_1 \\ D_2 \\ \vdots \\ D_p \end{bmatrix} = \begin{bmatrix} D_1\eta^1 & D_1\eta^2 & \cdots & D_1\eta^p \\ D_2\eta^1 & D_2\eta^2 & \cdots & D_2\eta^p \\ \vdots & \vdots & & \vdots \\ D_p\eta^1 & D_p\eta^2 & \cdots & D_p\eta^p \end{bmatrix} \begin{bmatrix} \mathcal{D}_1 \\ \mathcal{D}_2 \\ \vdots \\ \mathcal{D}_p \end{bmatrix}. \tag{16}$$

Since $J = [D_i \eta^j]$ is non-degenerate, Cramer's rule yields

$$\mathcal{D}_k \zeta = \frac{\det(D_i \eta^1 \mid \cdots \mid D_i \eta^{k-1} \mid D_i \zeta \mid D_i \eta^{k+1} \mid \cdots \mid D_i \eta^p)}{\det(D_i \eta^j)}. \tag{17}$$

$\square$

*Remark* A.3. We require that the differential invariants $\eta^1, \cdots, \eta^p$ has a nondegenerate horizontal Jacobian $[D_i \eta^j]$, which is a stronger condition than functional independence. Since the differential invariants are functions on the jet space, it is possible that a set of such functions is functionally independent, i.e., has a nondegenerate full Jacobian $[\partial_i \eta^j]$, where $i \in [q_n]$ indexes the jet space variables $(\mathbf{x}, u^{(n)})$, but has a lower-rank horizontal Jacobian. For example, consider $\eta^1 = u_x$ and $\eta^2 = u_y$. In the full Jacobian, $\partial \eta^j / \partial u_x$ and $\partial \eta^j / \partial u_y$ form the identity, so it has full rank. However, its horizontal Jacobian containing total derivatives is given by $\begin{bmatrix} u_{xx} & u_{xy} \\ u_{xy} & u_{yy} \end{bmatrix}$, which is not invertible on the subset of the jet space where $u_{xx} u_{yy} - u_{xy}^2 = 0$.

In practice, this non-degeneracy condition can be easily checked once we have the symbolic expressions of the $p$ differential invariants.

*Remark* A.4. When $p = 1$, Proposition A.2 is equivalent to the following (Prop. 2.53, Olver (1993)):

If $y = \eta(x, u^{(n)})$ and $w = \zeta(x, u^{(n)})$ are $n$-th order differential invariants of $G$, then $\frac{dw}{dy} \equiv \frac{D_x \zeta}{D_x \eta}$ is an $(n+1)$-th order differential invariant of $G$. Specifically, if $y = \eta(x, u)$ and $w = \zeta(x, u, u_x)$ form a complete set of functionally independent differential invariants of $\mathrm{pr}^{(1)} G$, the complete set of functionally independent differential invariants for $\mathrm{pr}^{(n)} G$ is then given by

$$y, w, dw/dy, ..., d^{n-1}w/dy^{n-1}. \tag{18}$$

## A.4 Examples of Computing Differential Invariants

*Example* A.5. Consider the group $SO(2)$ acting on $X \times U \simeq \mathbb{R}^2 \times \mathbb{R}$ by standard rotation in the 2D space of independent variable and trivial action on $U$, i.e. its infinitesimal generator given by $\mathbf{v} = y\partial_x - x\partial_y$.

First, we solve for a complete set of the ordinary and first-order invariants. By definition, the ordinary invariants $\eta = \eta(x, y, u)$ should satisfy $y\partial_x \eta - x\partial_y \eta = 0$. Since the vector field does not involve $u$, an immediate solution is $\eta = u$. On the othe hand, by method of characteristics, we convert the PDE to the characteristic equations $dx/ds = y, dy/ds = -x$. That is, the characteristics curves $(x(s), y(s))$ are just circles around origin. Because $\eta$ is constant along characteristic curves, it must be a function of $R^2 = x^2 + y^2$. Therefore, we pick the following two ordinary invariants: $\eta_1(x, y, u) = \frac{1}{2}(x^2 + y^2)$ and $\eta_2(x, y, u) = u$. (5) dictates how we construct higher-order invariants using these two functionally independent invariants and another arbitrary invariant. For notational convenience, we convert (5) to operators defined according to $\eta_2$ and $\eta_1$, respectively:

$$\mathcal{O}_1 = \frac{xD_y - yD_x}{xu_y - yu_x} \tag{19}$$

$$\mathcal{O}_2 = \frac{u_y D_x - u_x D_y}{xu_y - yu_x} \tag{20}$$

Then, we need to find another new differential invariant, because applying these operators on $\eta_1$ and $\eta_2$ leads to trivial results. Since $\eta_1$ and $\eta_2$ generate all ordinary (zeroth-order) invariants, we must look for the first-order invariants. To do this, note the prolonged vector field is given by

$$\mathrm{pr}^{(1)} \mathbf{v} = \mathbf{v} + u_y \partial_{u_x} - u_x \partial_{u_y} \tag{21}$$

Solving for $\mathrm{pr}^{(1)} \mathbf{v}$ gives two first-order invariants, $\zeta_1 = xu_y - yu_x$ and $\zeta_2 = xu_x + yu_y$. Note that the differential invariant $\zeta_1$ is exactly the common denominator in $\mathcal{O}_1$ and $\mathcal{O}_2$, so we can simplify $\mathcal{O}_1$ and $\mathcal{O}_2$ by

using only their numerators, i.e.

$$\mathcal{O}_1 = xD_y - yD_x \tag{22}$$
$$\mathcal{O}_2 = u_y D_x - u_x D_y \tag{23}$$

Note that $\mathcal{O}_2$ has first-order coefficients, which may complicate things in the subsequent calculation. Denoting the space of all continuous functions of the existing four invariants as $\mathcal{I} = \mathcal{C}(\eta_1, \eta_2, \zeta_1, \zeta_2)$, we can choose any new operator within the $\mathcal{I}$-module spanned by $\mathcal{O}_1$ and $\mathcal{O}_2$ that makes things easier. Specifically, we use the following operator

$$\tilde{\mathcal{O}}_2 = \frac{\zeta_2}{\zeta_1}\mathcal{O}_1 + \frac{2\eta_1}{\zeta_1}\mathcal{O}_2$$
$$= xD_x + yD_y \tag{24}$$

Then, we apply these operators to the first-order invariants, which raise the order by one and give us the second-order invariants. For example, applying $O_1$ to $\zeta_1$, we have

$$\begin{aligned}
\mathcal{O}_1\zeta_1 &= xD_y\zeta_1 - yD_x\zeta_1 \\
&= x(xu_{yy} - u_x - yu_{xy}) - y(u_y + xu_{xy} - yu_{xx}) \\
&= x^2 u_{yy} + y^2 u_{xx} - xu_x - yu_y - 2xyu_{xy}
\end{aligned} \tag{25}$$

Note that $\zeta_2 = xu_x + yu_y$ is a first-order invariant, so we can further remove it from the formula and get a simplified second-order invariant

$$\vartheta_1 = x^2 u_{yy} + y^2 u_{xx} - 2xyu_{xy} \tag{26}$$

Similarly, we compute $\mathcal{O}_1\zeta_2$, $\tilde{\mathcal{O}}_2\zeta_1$ and $\tilde{\mathcal{O}}_2\zeta_2$ and obtain the following, respectively:

$$\begin{aligned}
\vartheta_2 = \vartheta_3 &= \zeta_1 + xy(u_{yy} - u_{xx}) + (x^2 - y^2)u_{xy} \\
&\equiv xy(u_{yy} - u_{xx}) + (x^2 - y^2)u_{xy}
\end{aligned} \tag{27}$$
$$\begin{aligned}
\vartheta_4 &= \zeta_2 + x^2 u_{xx} + y^2 u_{yy} + 2xyu_{xy} \\
&\equiv x^2 u_{xx} + y^2 u_{yy} + 2xyu_{xy}
\end{aligned} \tag{28}$$

The above 8 invariants should form a complete set of second-order differential invariants of $\mathbf{v} = x\partial_y - y\partial_x$. To verify, note that the Laplacian $\Delta u = u_{xx} + u_{yy}$, which is a well-known rotational invariant, can be written in terms of these differential functions:

$$\begin{aligned}
\Delta u = u_{xx} + u_{yy} &= \frac{(x^2 + y^2)(u_{xx} + u_{yy})}{x^2 + y^2} \\
&= \frac{\vartheta_1 + \vartheta_4}{2\eta_1}
\end{aligned} \tag{29}$$

Another second-order rotational invariant, the trace of the squared Hessian matrix, $u_{xx}^2 + 2u_{xy}^2 + u_{yy}^2$, is recovered by

$$u_{xx}^2 + 2u_{xy}^2 + u_{yy}^2 = \frac{\vartheta_1^2 + 2\vartheta_2^2 + \vartheta_4^2}{4\eta_1^2} \tag{30}$$

On the other hand, these 8 invariants are apparently not functionally independent - note that $\vartheta_2 = \mathcal{O}_1\zeta_2$ and $\vartheta_3 = \tilde{\mathcal{O}}_2\zeta_1$ are the same. While this may be some coincidence, eventually it is not surprising because we would expect to see 3 functionally independent strictly second-order differential invariants instead of 4, since $(u_{xx}, u_{yy}, u_{xy}) \in U_2$ is only 3-dimensional.

*Example* A.6 (Scaling and translation). Consider the vector field $\mathbf{v}_1 = t\partial_t + ax\partial_x + bu\partial_u$. It generates the scaling symmetry $t \mapsto \lambda t, x \mapsto \lambda^a x, u \mapsto \lambda^b u$. The ordinary invariants of this symmetry are $t^b u^{-1}$ and $x^a u^{-1}$. The higher-order invariants are given by $\eta_{(\alpha,\beta)} = x^\alpha t^\beta u_{x^{(\alpha)} t^{(\beta)}} u^{-1}$, where $\alpha$ and $\beta$ denote the orders of partial derivatives w.r.t $t$ and $x$, e.g. $u_{x^{(2)} t^{(1)}} := u_{xxt}$.

Besides the scaling symmetry, we can consider other common symmetries simultaneously, e.g. translation symmetries in both space and time, $\mathbf{v}_2 = \partial_x$ and $\mathbf{v}_3 = \partial_t$. These symmetries, along with the scaling symmetry $\mathbf{v}_1$, span a three-dimensional symmetry group. There are no ordinary invariants due to the translation symmetries. A convenient maximal set of functionally independent differential invariants is given by

$$\eta_{(\alpha,\beta)} = u_{x^{(\alpha)} t^{(\beta)}} u_x^{\frac{b - a\alpha - \beta}{a - b}}, \ \alpha \geq 0, \beta \geq 0. \tag{31}$$

## A.5 Proof of Proposition 4.4

Proposition 4.4, restated below, aligns our symmetry constraint into the SINDy framework and results in a set of constraints on the SINDy parameters.

**Proposition A.7.** *Let* $\boldsymbol{\ell}(\mathbf{x}, \mathbf{u}^{(n)}) = W\boldsymbol{\theta}(\mathbf{x}, \mathbf{u}^{(n)})$ *be a system of* $q$ *differential equations admitting a symmetry group* $G$, *where* $\mathbf{x} \in \mathbb{R}^p$, $\mathbf{u} \in \mathbb{R}^q$, $\boldsymbol{\theta} \in \mathbb{R}^m$. *Assume there exist some nth-order invariants of* $G$, $\eta_0^{1:q}$ *and* $\eta^{1:K}$, *s.t. (1) the system of equations can be expressed as* $\boldsymbol{\eta}_0 = W'\boldsymbol{\theta}'(\boldsymbol{\eta})$, *where* $\boldsymbol{\eta}_0 = [\eta_0^{1:q}]$ *and* $\boldsymbol{\eta} = [\eta^{1:K}]$, *and (2)* $\eta_0^i = T^{ijk}\theta^k\ell^j$ *and* $(\theta')^i = S^{ij}\theta^j$, *for some functions* $\boldsymbol{\theta}'(\boldsymbol{\eta})$ *and constant tensors* $W'$, $T$ *and* $S$. *Then, the space of all possible* $W$ *is a linear subspace of* $\mathbb{R}^{q \times m}$.

*Proof.* (Note: In this proof, we do not distinguish between superscripts and subscripts. All are used for tensor indices, not partial derivatives.)

For simplicity, we omit the dependency of functions and write

$$\ell^i = W^{ij}\theta^j. \tag{32}$$

Combining the conditions about the differential invariants, we know that the equation can be equivalently expressed as

$$T^{ijk}\theta^k\ell^j = (W')^{ij}S^{jk}\theta^k \tag{33}$$

for some $W' \in \mathbb{R}^{q \times m'}$, where $m'$ is the number of invariant functions in $\boldsymbol{\theta}'$.

Substituting (32) into (33) and rearranging the indices, the principle of symmetry invariants then translates to the following constraint on $W$: there exists some $W' \in \mathbb{R}^{q \times m'}$ s.t.

$$T_i{}^{rk}\theta_k W_r{}^l\theta_l = (W')_i{}^k S_k{}^j\theta_j, \forall\mathbf{x}, \mathbf{u}^{(n)}. \tag{34}$$

To solve for $W$, we first eliminate the dependency on the variables $\mathbf{x}$ and $\mathbf{u}^{(n)}$ from the equation. We adopt a procedure similar to Yang et al. (2024). Denote $\mathbf{z} = (\mathbf{x}, \mathbf{u}^{(n)})$. Define a functional $M_{\boldsymbol{\theta}}$ as mapping a function to its coordinate in the function space spanned by $\boldsymbol{\theta}$, i.e. $M_{\boldsymbol{\theta}} : (\mathbf{z} \mapsto c^j\theta_j(\mathbf{z})) \mapsto (c^1, c^2, \cdots, c^m)$. Before we proceed, note that the LHS of (34) contains the products of functions $\theta_k(\mathbf{z})\theta_l(\mathbf{z})$, which may or may not be included in the original function library $\boldsymbol{\theta}$. Therefore, we denote $\tilde{\boldsymbol{\theta}}(\mathbf{z}) = [\boldsymbol{\theta}(\mathbf{z}) \| \{\theta_k\theta_l \notin \boldsymbol{\theta}\}]$ as the collection of all library functions $\theta_k$ and all their products $\theta_k\theta_l$. The invariant functions $\boldsymbol{\theta}'(\boldsymbol{\eta})$ can also be rewritten in terms of the prolonged library: $\boldsymbol{\theta}'(\boldsymbol{\eta}) = \tilde{S}\tilde{\boldsymbol{\theta}}$, where $\tilde{S}_{1:m} = S$.

Then, applying $M_{\tilde{\boldsymbol{\theta}}}$ to (34), we have

$$M_{\tilde{\boldsymbol{\theta}}}(T_i{}^{rk}\theta_k W_r{}^l\theta_l) = (W')_i{}^k \tilde{S}_k{}^j. \tag{35}$$

Further expanding the LHS, we have

$$T_i{}^{rk}W_r{}^l\Gamma_{kl}{}^j = (W')_i{}^k \tilde{S}_k{}^j, \tag{36}$$

where $\Gamma$ satisfies $\theta_k \theta_l = \Gamma_{kl}{}^j \tilde{\theta}_j$. In other words, the rows of the LHS fall in the row space of $\tilde{S}$. Let $\tilde{S}^\perp$ be the basis matrix for the null space of $\tilde{S}$, i.e. $\tilde{S}\tilde{S}^\perp = 0$, we have

$$T_i{}^{rk} W_r{}^l \Gamma_{kl}{}^j (\tilde{S}^\perp)_{js} = 0, \tag{37}$$

suggesting that $W$ must lie in a linear subspace of $\mathbb{R}^{q \times m}$.

$\square$

*Remark* A.8. In practice, to solve for (37), we first rearrange (37) into $M\mathrm{vec}(W) = 0$, where $M$ has shape $(\tilde{S}.\mathrm{shape}[2] \times q, \ q \times m)$. Then, we perform SVD on $M$ and apply a threshold of $10^{-6}$ to the singular values. The right singular vectors corresponding to the singular values smaller than the threshold then form a basis of the linear subspace $\mathrm{vec}(W)$ lies in.

# B   Implementation Details

This section discusses some detailed considerations in implementing the sparse regression-based methods described in Section 4.3 and 4.4. Contents include:

- Appendix B.1: An algorithmic description of direct sparse regression with symmetry invariants.

- Appendix B.2: Converting the symmetry invariant condition as linear constraints on the sparse regression parameters.

- Appendix B.3: Using differential invariants in weak SINDy via the linear constraints, as well as other considerations.

## B.1   Direct Sparse Regression With Symmetry Invariants

The first approach to enforcing symmetry in sparse regression, as discussed in Section 4.3, is to directly use the symmetry invariants as the variables and their functions specified by a function library as the RHS features. Similar to Algorithm 1 for general symbolic regression methods, we provide a detailed algorithm for sparse regression below. Following the setup from SINDy, we aim to discover a system of $q$ differential equations for $q$ dependent variables.

---

**Algorithm 2** Sparse regression with symmetry invariants

---

**Require:** PDE order $n$, dataset $\{\mathbf{z}^i = (\mathbf{x}^i, (\mathbf{u}^{(n)})^i) \in M^{(n)}\}_{i=1}^{N_D}$, SINDy LHS $\boldsymbol{\ell}$, SINDy function library $\{\theta^j\}$, infinitesimal generators of the symmetry group $\mathcal{B} = \{\mathbf{v}_a\}$.
**Ensure:** A PDE system admitting the given symmetry group.
 Compute the symmetry invariants of $\mathcal{B}$ up to $n$th-order: $\eta^1, \cdots, \eta^K$. {Prop. 4.3}
 Choose an invariant function $\eta^{k_i}$ s.t. $\partial \eta^{k_i} / \partial \ell^i \neq 0$ for SINDy LHS component $\ell^i$.
 Let $\boldsymbol{\eta}_0 = [\eta^{k_1}, ..., \eta^{k_q}]^T$ and $\boldsymbol{\eta}$ denote the column vector containing the remaining $K - q$ invariants.
 Instantiate the sparse regression model as $\boldsymbol{\eta}_0 = W\boldsymbol{\theta}(\boldsymbol{\eta})$.
 Optimize $W$ with the SINDy objective: $\sum_i \|\boldsymbol{\eta}_0(\mathbf{z}^i) - W\boldsymbol{\theta}(\boldsymbol{\eta}(\mathbf{z}^i))\|^2 + \lambda \|W\|_0$.
 **return** $\boldsymbol{\eta}_0 = W\boldsymbol{\theta}(\boldsymbol{\eta})$. {Optionally, expand all $\eta^j$ in terms of original variables $\mathbf{z}$.}

---

The configuration from the original SINDy model, i.e., the LHS $\boldsymbol{\ell}$ and the function library $\{\theta^j\}$, are used to construct a new equation model in terms of the invariants. It should be noted that the functions in the SINDy function library does not specify their input variables. For example, in the PySINDy (Kaptanoglu et al., 2022) implementation, a function $\theta$ is provided in a lambda format `lambda x, y:  x * y`. Thus, $\theta$ can be applied to both the original variables, e.g. $\theta(z_1, z_2) = z_1 z_2$, and the invariant functions, e.g. $\theta(\eta_1, \eta_2) = \eta_1 \eta_2$.

## B.2   Symmetry Invariant Condition as Linear Constraints

Instead of directly using the invariant functions $\eta$ as the features and labels for regression, we can derive a set of linear constraints from the fact that the equation can be rewritten in terms of invariant functions. As

shown in Appendix A.5, a basis $Q$ of the constrained parameter space can be obtained from the right singular vectors of a constraint matrix $M$. We rearrange $Q$ to a tensor of shape $(r, q, m)$, where $r$ is the dimension of the constrained parameter space, and $(q, m)$ is the original shape of the parameter matrix $W$. Then, we can parameterize $W$ by $W^{jk} = Q^{ijk}\beta^i$, where $\beta$ is the learnable parameter, and discover the equation using the original SINDy objective as described in Section 3.2.

In practice, we observe that the basis $Q$ obtained from SVD on (37) is not sparse. Indeed, SVD does not inherently encourage sparsity in the singular vectors. As mentioned in Section 4.3, the lack of sparsity can pose a problem when we perform sequential thresholding in sparse regression. Therefore, after performing SVD, we apply a Sparse PCA to $Q$ to obtain a sparsified basis, also of shape $(r, q, m)$:

```
spca = SparsePCA(n_components=r)
spca.fit(Q.reshape(r, q*m))
Q_sparse = spca.components.reshape(r,q,m)
```

Figure 6 shows an example of the original basis solved from SVD (top $7 \times 2$ grid) and the sparsified basis using sparse PCA (bottom $7 \times 2$ grid). This is used in our experiment on the reaction-diffusion system (8).

### B.3  Using Differential Invariants in Weak SINDy

In this subsection, we discuss the formulation of weak SINDy and how to implement our strategy of using differential invariants within the weak SINDy framework. To maintain a similar notation to the original works on weak SINDy (Messenger & Bortz, 2021a;b), we use $D_{\alpha_s}$ to denote partial derivative operators, where $\alpha_s = (s_1, s_2, ..., s_p)$ is a multi-index, instead of using subscripts for partial derivatives. Thus, we no longer strictly differentiate subscripts and superscripts–both can be used for indexing lists, vectors, etc.

Given a differential equation in the form

$$D_{\alpha_0} u = \sum_{s,j} W_{sj} D_{\alpha_s} f_j(u), \tag{38}$$

we can perform integration by parts (i.e., divergence theorem) to move the derivatives from $u$ to some analytic test function and thus bypass the need to estimate derivatives numerically. First, we multiply both sides of (38) by a test function $\phi$ with compact support $\mathcal{B} \subset X$ and integrate over the spacetime domain:

$$\int_X D_{\alpha_0} u(\mathbf{x})\phi(\mathbf{x})d\mathbf{x} = \sum_{s,j} W_{sj} \int_X D_{\alpha_s} f_j(u(\mathbf{x}))\phi(\mathbf{x})d\mathbf{x} \tag{39}$$

WLOG, assume that $s_1 \neq 0$, and denote $\alpha_{s'} = (s_1 - 1, s_2, ..., s_p)$. Then, each term in the RHS can be integrated by parts as

$$\begin{aligned}
\int_X D_{\alpha_s} f_j(u(\mathbf{x}))\phi(\mathbf{x})d\mathbf{x} &= \int_{\mathcal{B}} D_{\alpha_s} f_j(u(\mathbf{x}))\phi(\mathbf{x})d\mathbf{x} \\
&= -\int_{\mathcal{B}} D_{\alpha_{s'}} f_j(u(\mathbf{x}))D_1\phi(\mathbf{x})d\mathbf{x} + \int_{\partial\mathcal{B}} \nu_1 D_{\alpha_{s'}} f_j(u(\mathbf{x}))\phi(\mathbf{x})d\mathbf{x} \\
&= -\int_{\mathcal{B}} D_{\alpha_{s'}} f_j(u(\mathbf{x}))D_1\phi(\mathbf{x})d\mathbf{x}, \tag{40}
\end{aligned}$$

where $D_1$ denotes the partial derivative operator w.r.t the first independent variable, and $\nu_1$ is the first component of the unit outward normal vector.

Repeating this process until all the derivative operations move from $f_j(u)$ to the test function $\phi$, we have

$$\int_X D_{\alpha_s} f_j(u(\mathbf{x}))\phi(\mathbf{x})d\mathbf{x} = (-1)^{|\alpha_s|} \int_X f_j(u(\mathbf{x}))D_{\alpha_s}\phi(\mathbf{x})d\mathbf{x} \tag{41}$$

Similarly for the LHS:

$$\int_X D_{\alpha_0} u(\mathbf{x})\phi(\mathbf{x})d\mathbf{x} = (-1)^{|\alpha_0|} \int_X u(\mathbf{x})D_{\alpha_0}\phi(\mathbf{x})d\mathbf{x} \tag{42}$$

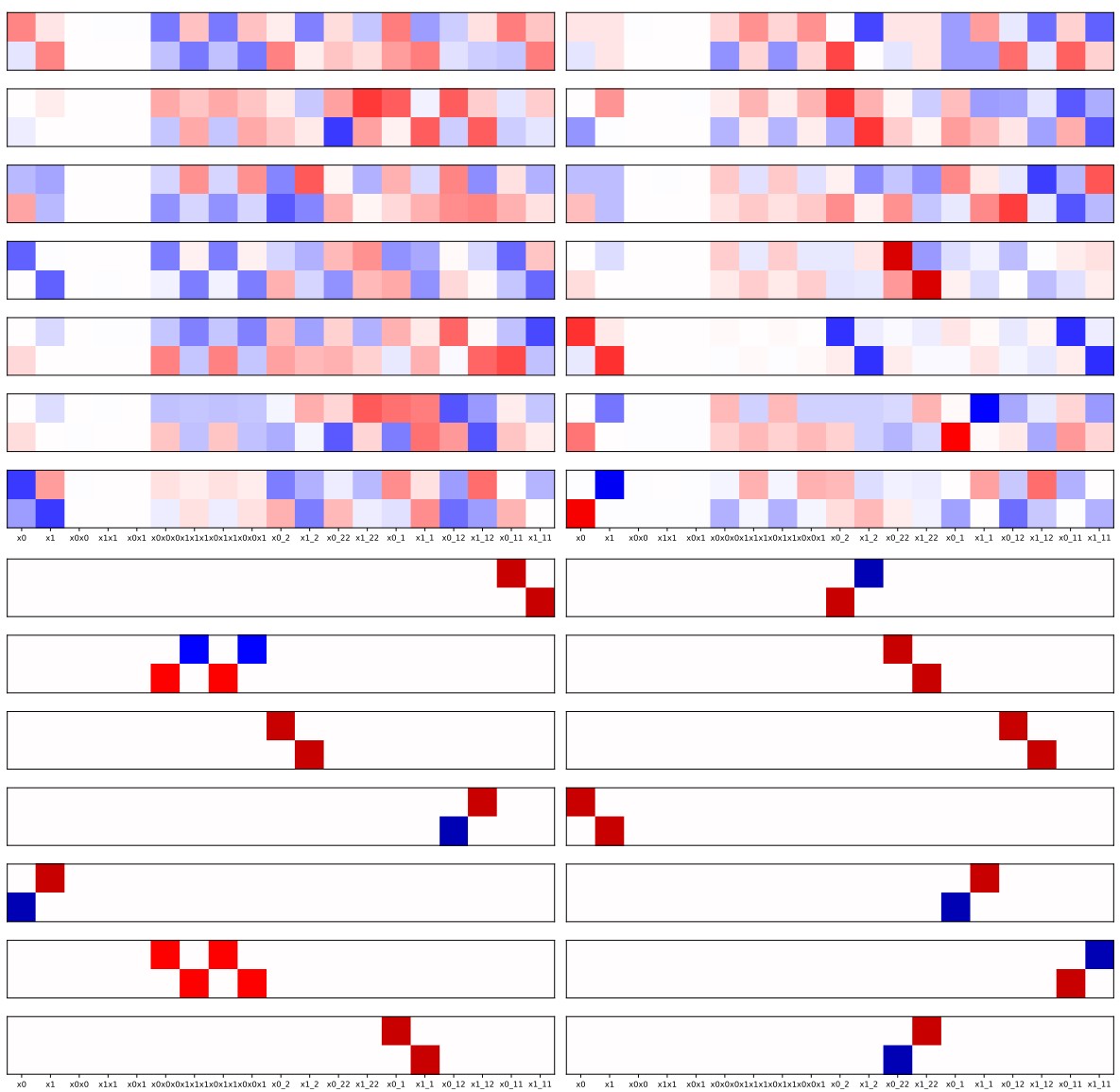

Figure 6: Basis for the SINDy parameter subspace that preserves SO(2) symmetry $\mathbf{v} = -v\partial_u + u\partial_v$. The SINDy parameter $W$ has dimension $2 \times 19$. The two rows correspond to the two equations with $u_t$ and $v_t$ as the LHSs. The RHS contains 19 features, including all monomials of $u, v$ up to degree 3 and their spatial derivatives up to order 2. The set of symmetry invariants used to compute the basis is given by $\{t, x, y, u^2 + v^2\} \bigcup \{\mathbf{u} \cdot \mathbf{u}_\mu\} \bigcup \{\mathbf{u}^\perp \cdot \mathbf{u}_\mu\}$, where $\mathbf{u} = (u, v)^T$ and $\mu$ is a multiindex of $t, x, y$ with order no more than 2. The top $7 \times 2$ grid displays the original basis solved from SVD, and the bottom $7 \times 2$ grid displays the sparsified basis.

The final optimization problem is to solve for $\mathbf{b} = \mathbf{Gw}$, where $\mathbf{w}$ is the vectorized coefficient matrix $W$, and each row in $\mathbf{b}$ and $\mathbf{G}$ is given by computing the integrals in (41) and (42) against a single test function. The number of rows equals the number of different test functions used.

**Direct integration of symmetry via linear constraints**   As we have discussed in Appendix B.2, we can enforce symmetry by converting it to a set of linear constraints on the parameter $W$. With this approach, we can directly incorporate symmetry in weak SINDy. Specifically, we just parameterize $W$ as in terms of a precomputed basis $Q$ and a trainable vector $\beta$ and directly substitute this parameterization of $W$ into the optimization problem of weak SINDy. We adopt this strategy in our experiments concerning weak SINDy.

**Expressing the equations with differential invariants**   The above approach is only possible when the conditions in Proposition 4.4 about the selected set of symmetry invariants hold. We should note that it is not always possible to find a set of invariants so that the symmetry condition can be converted to linear constraints on the parameter $W$ via the procedure in the proof of Proposition 4.4. One may ask the following question: can we simply express the equations in terms of differential invariants and apply weak SINDy, similar to Algorithm 2 for the original SINDy formulation? Here, we do not provide a definite conclusion for this question, but only discuss several cases where directly using differential invariants in equations might succeed or fail in weak SINDy.

To adapt to the weak SINDy formulation (38), it is more helpful to consider the symmetry invariants as generated by some fundamental invariants and some invariant differential operators, instead of specifying a complete set of differential invariants for every order. Concretely, there exists a set of invariant differential operators $\{\mathcal{O}_j\}$ and a set of fundamental differential invariants $\mathbf{I} = \{\eta_k\}$ s.t. every differential invariant can be written as $\mathcal{O}_{j_1}...\mathcal{O}_{j_n}\eta_k$. For the SO(2) symmetry group in Example A.5, one possible choice is

$$\eta_1 = \frac{1}{2}(x^2 + y^2), \ \eta_2 = u, \ \mathcal{O}_1 = xD_y - yD_x, \ \mathcal{O}_2 = xD_x + yD_y. \tag{43}$$

We can compose these generating invariant operators to obtain a full library of eligible differential operators up to some order, denoted $\mathbf{D} = \{\mathcal{D}_j\}$. The exact compositions can vary and we can choose the most convenient one for subsequent calculations. For the above SO(2) example, for up to second-order differential operators, we can choose $\{\mathcal{O}_1, \mathcal{O}_2, \mathcal{O}_1^2, \mathcal{O}_2^2, \frac{2}{\eta_1}(\mathcal{O}_1^2 + \mathcal{O}_2^2)\}$. Note the last operator is exactly the Laplacian.

Then, the complete set of eligible terms (respecting the symmetry) in the equation is $\{\mathcal{D}_j\eta_k : \mathcal{D}_j \in \mathbf{D}, \eta_k \in \mathbf{I}\}$. If we assume, as in SINDy, that the governing equation can be written in linear combination of these symmetry invariants, then we can assign a weight for each $\mathcal{D}_j\eta_k$ and form a coefficient matrix $W = [W_{jk}]$. That is,

$$\mathcal{D}_{j_0}\eta_{k_0} = \sum_{(j,k)\neq(j_0,k_0)} W_{jk}\mathcal{D}_j\eta_k. \tag{44}$$

Then, multiplying each side by a test function $\phi(\mathbf{x})$, we have

$$\int_X \mathcal{D}_{j_0}\eta_{k_0}\phi(\mathbf{x})d\mathbf{x} = \sum_{(j,k)\neq(j_0,k_0)} W_{jk} \int_X \mathcal{D}_j\eta_k\phi(\mathbf{x})d\mathbf{x}. \tag{45}$$

The question then boils down to whether we can apply the technique of integration by parts similarly to this set of differential operators and differential functions, since the original algorithm only deals with partial derivative operators $D_{\alpha_s}$ and ordinary functions $f_j(u)$.

To check this, let us explicitly write out the dependency of these operators and fundamental invariants.

**Case 1** A relatively simple case is when all invariant operators take the form $\mathcal{D}_j = \sum_s a_s(\mathbf{x})D_{\alpha_s}$ and $\eta_k = \eta_k(\mathbf{x}, u(\mathbf{x}))$. Each term in the RHS of (45) can be expanded as

$$
\int_X \mathcal{D}_j \eta_k \phi(\mathbf{x}) d\mathbf{x} = \sum_s \int_X a_s(\mathbf{x}) D_{\alpha_s} \eta_k(\mathbf{x}, u(\mathbf{x})) \phi(\mathbf{x}) d\mathbf{x}
$$
$$
= \sum_s (-1)^{|\alpha_s|} \int_X \eta_k(\mathbf{x}, u(\mathbf{x})) D_{\alpha_s}[a_s(\mathbf{x})\phi(\mathbf{x})] d\mathbf{x} \tag{46}
$$

Evaluating (46) does not require estimating partial derivatives of $u$. Therefore, weak SINDy can be applied to this case quite straightforwardly.

**Case 2** However, it is not always possible to have all $\mathcal{D}_j$ as classical linear differential operators and all $\eta_k$ as ordinary functions. For instance, in Example A.6, there are no ordinary symmetry invariants due to the constraint of translation symmetry.

If we still have linear operators $\mathcal{D}_j = \sum_s a_s(\mathbf{x})D_{\alpha_s}$, but on the other hand we have differential functions $\eta_k = \eta_k(\mathbf{x}, u^{(n)})$, we can still perform integration by parts as in (46), but the final result becomes

$$
\sum_s (-1)^{|\alpha_s|} \int_X \eta_k(\mathbf{x}, u^{(n)}) D_{\alpha_s}[a_s(\mathbf{x})\phi(\mathbf{x})] d\mathbf{x}, \tag{47}
$$

meaning we still have to evaluate whatever partial derivatives remain in $\eta_k$. It is possible that we can decrease the order of partial derivatives compared to vanilla sparse regression, but we cannot eliminate all partial derivatives compared to Weak SINDy without any symmetry information.

**Case 3** The most challenging case is when the invariant differential operators explicitly involve the partial derivative, such as $\mathcal{D}_j = \sum_s a_s(\mathbf{x}, u^{(n)})D_{\alpha_s}$. Then, similar to (47), integration by parts yields:

$$
\sum_s (-1)^{|\alpha_s|} \int_X \eta_k(\mathbf{x}, u^{(n)}) D_{\alpha_s}[a_s(\mathbf{x}, u^{(n)})\phi(\mathbf{x})] d\mathbf{x}. \tag{48}
$$

In this case, we still need to compute the partial derivatives, not only those in $\eta_k$, but also those arising from $a_s$ and $D_{\alpha_s}(a_s)$. The latter might involve higher-order derivatives and the benefit of using the weak formulation may further diminish.

## C Additional Experiment Results

Contents of this section include:

- Appendix C.1: Extended results in Table 1 with confidence intervals for the prediction error metric over 100 runs.

- Appendix C.2: Results for some variants of the sparse regression models considered in Table 1.

- Appendix C.3: Results for the symmetry-breaking reaction-diffusion systems with auto-tuned WSINDy parameters, complementing Figure 4.

- Appendix C.4: Results on a 3D PDE system with SO(3) spatial symmetry.

- Appendix C.5: Results for the D-CIPHER (Kacprzyk et al., 2023) baseline and our method applied to D-CIPHER on the Darcy flow dataset.

- Appendix C.6: Samples of equations discovered by different methods.

- Appendix C.7: Visualized prediction errors of equations discovered by different methods.

## C.1 Results in Table 1 With Error Estimates

Table 3: Extended results in Table 1 with confidence intervals for the prediction error metric over 100 runs. Each table entry is formatted as median [25% quantile, 75% quantile].

| Method | | Boussinesq (6) | Darcy flow (7) | Reaction-diffusion (8) |
|---|---|---|---|---|
| Sparse Regression | PySINDy | 0.373 [0.367, 0.380] | - | 0.021 [0.020, 0.022] |
| | SI | **0.098** [0.098, 0.098] | - | **0.008** [0.007, 0.013] |
| Genetic Programming | PySR | 0.098 [0.098, 0.098] | 0.114 [0.089, 0.169] | - |
| | SI | 0.098 [0.098, 0.098] | **0.051** [0.031, 0.053] | **0.023** [0.015, 0.036] |
| Transformer | E2E | 0.132 [0.109, 0.322] | - | - |
| | SI | **0.104** [0.100, 0.109] | - | - |

## C.2 Variant Sparse Regression Models

Table 4: Results of sparse regression models on the Boussinesq equation and the reaction-diffusion system. $\mathcal{C}$ stands for complexity, i.e., the dimensionality of the parameter space. SP stands for success probability. The PySINDy and SI rows present the same results as the corresponding rows in Table 1.

| Method | Boussinesq (6) | | Reaction-diffusion (8) | |
|---|---|---|---|---|
| | $\mathcal{C} \downarrow$ | SP $\uparrow$ | $\mathcal{C} \downarrow$ | SP $\uparrow$ |
| PySINDy | 15 | 0.00 | 38 | 0.53 |
| PySINDy* | 21 | **1.00** | 468 | 0.00 |
| PySINDy** | 15 | **1.00** | 198 | 0.00 |
| SI | 15 | **1.00** | 28 | 0.54 |
| SI-aligned | - | - | **14** | **0.56** |

PySINDy (de Silva et al., 2020; Kaptanoglu et al., 2022) constructs its library $\boldsymbol{\theta}$ from a list of variables and derivatives, $[\mathbf{u} \,||\, \mathbf{u}_\alpha]$ ($|\alpha| > 0$) and a set of scalar functions specified in lambda format. For example, to include up to quadratic monomial terms in the library, we can specify the following functions: $x \mapsto x$ and $(x, y) \mapsto xy$. However, their original implementation does not allow these functions to be applied to partial derivative terms. As a result, terms such as $u_x^2$ cannot be modeled. This leads to its failure to discover the Boussinesq equation (6), as we have shown in Table 1.

We modify the implementation and include an additional set of results with different libraries, denoted as PySINDy* in Table 4. The PySINDy* model supports a wider range of library functions, including functions of partial derivatives, e.g., $u_x^2$. Further more, we notice that the PySINDy* library while comprehensive, contains many redundant terms, such as interactions between derivatives like $u_x u_{xx}$. Therefore, we implement another library, denoted PySINDy**, where functions such as $(x, y) \mapsto xy$ are only applied when their arguments do not contain at least two different partial derivatives. Therefore, PySINDy** library still includes all the necessary terms to recover the Boussinesq equation but becomes much more compact. A complete description of the hypothesis spaces of different sparse regression-based methods is available in Appendix E.5.

As Table 4 shows, both PySINDy* and PySINDy** succeed in the Boussinesq equation. However, they fails in the reaction-diffusion system because their parameter spaces become too large due to a higher-dimensional total space $X \times U \simeq \mathbb{R}^2 \times \mathbb{R}^2$. Even with the more compact PySINDy**, there are still 198 possible terms for the reaction-diffusion system, and the algorithm never succeeded in 100 runs. This augments the point that SINDy's success relies on an appropriate choice of function library. If the library is too small to contain all the terms appearing in the equation of interest, the discovery is sure to fail. If the library is too large, the optimization problem becomes more difficult in the high-dimensional parameter space. On the other hand, by introducing the inductive bias of symmetry, our method automatically identifies a proper function library

that contains all the necessary terms for a PDE with a specific symmetry group, but not other redundant terms.

We include another model in Table 4, SI-aligned, where we derive a set of linear constraints on the sparse regression parameters from the fact that the equations can be expressed in terms of symmetry invariants. In this way, we still optimize the original parameters (though in a constrained subspace) as in the base SINDy model without symmetry, effectively "aligning" the hypotheses about equations from symmetry and the base SINDy model. This method is discussed in detail in Section 4.3 and Appendix B.2. We should also note that this method is mainly developed for incorporating the symmetry constraints into the weak formulation of SINDy. However, it is perfectly acceptable to implement it in the original formulation of SINDy, so we provide its results in Table 4 for reference.

For the reaction-diffusion system, SI-aligned has a 14-dimensional parameter space. The basis for its parameter space is visualized in Figure 6. It achieves a slightly higher success probability than SI (regression with symmetry invariants) and PySINDy (without symmetry information). We do not apply SI-aligned to the Boussinesq equation, because it is not necessary to align the hypotheses from SINDy and symmetry in this case. We can readily convert any equation discovered from SI (regression with symmetry invariants) by multiplying both sides by $u_x^2$.

We note that the results on the reaction-diffusion system in Table 4 are for models with the original SINDy formulation, in contrast to the weak SINDy formulation used in Figure 4. Therefore, the results in Figure 4 should not be directly compared to those in Table 1 and Table 4.

## C.3 Parameter Selection for Weak SINDy (WSINDy)

In Section 5.4, we have mentioned that different methods for selecting the test functions used in WSINDy have a significant impact on the equation discovery outcome. As is shown in Figure 4 (left), the auto-tuned test function parameters from the data result in better performance for baseline WSINDy (no symmetry constraint) in all noise levels considered, and for our symmetry-constrained method under relatively low noise levels.

In Figure 4 (center & right), we have used the default test function parameters from PySINDy for the variants of the reaction-diffusion system with imperfect symmetry. We find that the default parameters result in higher success probabilities of finding the correct equations in these cases. However, for comparison, we also show the results of the same experiments using the auto-selected test function parameters in Figure 7.

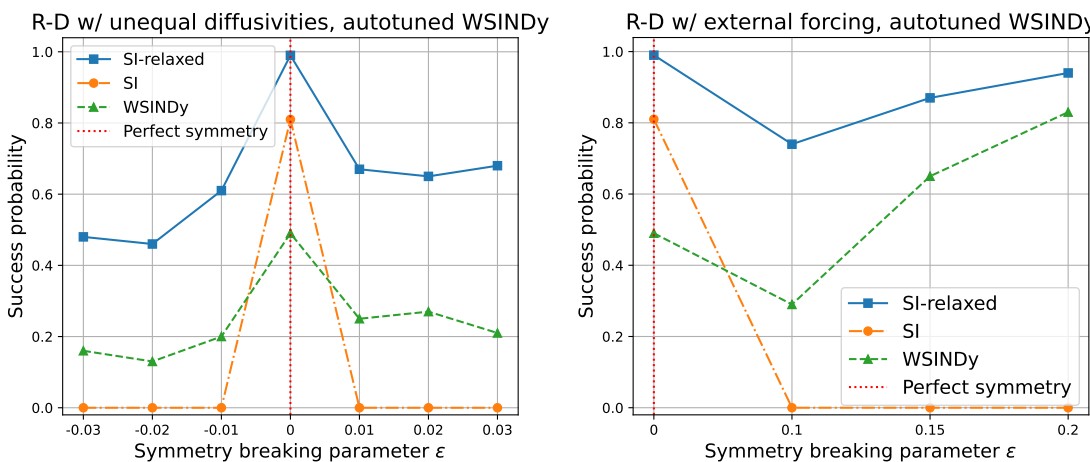

Figure 7: Success probabilities on the reaction-diffusion system with unequal diffusivities (left) and external forcing (right) with auto-tuned WSINDy parameters.

With the auto-tuning of test function parameters following the procedure in Messenger & Bortz (2021a), the relaxed version of our method (SI-relaxed) still has superior performance over the baseline WSINDy in both

symmetry-breaking scenarios. Notably, as the auto-tuning procedure typically suggests more test functions with higher polynomial degrees than the default setup, the performance of SI-relaxed is even better than its performance shown in Figure 4. However, the baseline WSINDy does not benefit from this selection strategy as much, while the version of our method that enforces hard constraints fails to recover any equation with imperfect symmetry.

### C.4 Equations with More Spatial Dimensions

Our previous examples focus on equations with 2 spatial dimensions at most. In this subsection, we test whether our method works well for discovering equations with higher spatial dimensionality. We consider a 3D isotropic reaction-diffusion system governed by

$$u_t = 0.2\Delta u + u - u^3, \tag{49}$$

where $u$ is a scalar field, and the nonlinear reaction term is given by $u - u^3$.

For data generation, we use a random perturbation around $u = 0$. Then, similar to the 2D reaction-diffusion system (8), we simulate the solution from this initial condition using an FFT-based spectral method with RK45 solver and periodic boundary conditions. We use 32 grid points per dimension and simulate up to $T = 10$, recording the solution at every $\Delta t = 0.1$. After simulation, we add a 1% noise to the trajectory and compute the derivatives from the noisy trajectory. We then split the data along the time dimension, using $t \in [0, 8)$ for training and $t \in [8, 10)$ for testing.

Table 5: Results on the 3D reaction-diffusion dataset over 100 runs for each method. The prediction error (PE) column is formatted as median [25% quantile, 75% quantile].

| Method | | $\mathcal{C} \downarrow$ | SP $\uparrow$ | PE $\downarrow$ |
|---|---|---|---|---|
| Sparse | PySINDy | 286 | 0.51 | 0.216 [0.207, 0.977] |
| regression | SI | **84** | **1.00** | **0.208** [0.204, 0.210] |
| Genetic | PySR | 10 | 0.00 | 1.228 [1.227, 1.230] |
| programming | SI | **6** | **0.20** | 1.227 [0.324, 1.229] |

The equation admits an SO(3) spatial symmetry. We use a set of SO(3) invariants up to second order given by $\mathcal{I}_{\text{RD-3D}} = \{u, (\nabla u)^2, \Delta u, (\nabla u)^T H \nabla u, \text{tr}(H^2), \det(H)\}$, where $H$ denotes the Hessian matrix $H_{ij} = u_{ij}$, and tr and det stands for matrix trace and determinant, respectively. Note that a complete set of invariants would also include terms with independent variables and time derivatives, e.g., $\{u_t, u_{tt}, \|\mathbf{x}\|_2^2, \mathbf{x}\cdot\nabla u, \mathbf{x}^T H \mathbf{x}, \mathbf{x}^T H \nabla u\}$. However, the standard setup of SINDy excludes time derivatives and independent variables in the RHS features. To align with the SINDy setup, we only use those 6 invariants in $\mathcal{I}_{\text{RD-3D}}$. Similarly, for genetic programming, even though it is possible to also include the spatial variables, we only use the standard coordinate in the 10-dimensional jet space (excluding $t$): $\{u, u_x, u_y, u_z, u_{xx}, u_{xy}, ..., u_{yz}, u_{zz}\}$. Finally, for the SINDy function library, we include all monomials up to degree 3 of all variables and derivatives.

Table 5 shows the results for sparse regression and GP-based methods on this 3D dataset. Our symmetry-invariant-based method (SI), when applied to SINDy, always discovers the correct functional form of the RHS of (49). In comparison, SINDy with regular variables only succeeds in about half of the 100 runs. Also, the complexity ($\mathcal{C}$), which stands for the total number of feature terms for SINDy-based methods, is much lower with our method. As such, we comment that our method's advantage of reducing the equation space complexity becomes more obvious in these higher-dimensional examples.

On the other hand, genetic programming proves to be a less favorable base SR algorithm in this case due to its overly large search space. PySR with standard jet variables completely fails to recover the correct equation. Our method uses the invariant set $\mathcal{I}_{\text{RD-3D}}$ as PySR input features and achieves a success probability (SP) of 20%. Finally, the symbolic transformer, which we used in the main experiments in Table 1 (E2E), has 0 success probability in this dataset, with either regular variables or symmetry invariants. We therefore do not report its results in Table 5. Again, we comment that the failure of the symbolic transformer might be because the data distribution from the PDE solution is largely different from its pretraining dataset.

## C.5 Comparison with D-CIPHER

A main advantage of our proposed method is its compatibility with various algorithms for symbolic regression. In the main experiments in Section 5.3 in Section 5.4, we have shown that our method works well with SINDy (Brunton et al., 2016), weak SINDy (Messenger & Bortz, 2021a), genetic programming (Cranmer, 2023), and symbolic transformer (Kamienny et al., 2022). To further demonstrate this advantage, we include another base algorithm for symbolic regression, D-CIPHER (Kacprzyk et al., 2023), in this section.

Similar to weak SINDy, D-CIPHER (Kacprzyk et al., 2023) uses a variational objective for equation discovery. It defines the extended derivative as

$$\mathcal{E}[\mathbf{u}](\mathbf{x}) = a(\mathbf{x})\partial_\alpha h(\mathbf{x}, \mathbf{u}),$$

where $a$ and $h$ are some functions and $\alpha$ is a multi-index indicating partial derivatives. Then, a library $\{\mathcal{E}^s\}_{s=1}^S$ of such extended derivatives is specified by the user by providing $S$ triples of $(a, \alpha, h)$. The algorithm then optimizes for a coefficient $\boldsymbol{\beta} \in \mathbb{R}^S$ and a symbolic function $g(\mathbf{x}, \mathbf{u})$ under a variational loss and outputs the equation

$$\sum_{s=1}^S \beta^s \mathcal{E}^s[\mathbf{u}](\mathbf{x}) = g(\mathbf{x}, \mathbf{u}).$$

As discussed in Appendix B.3, our approach can be directly used in D-CIPHER to enforce symmetry. We demonstrate this with an experiment on the Darcy flow dataset with SO(2) symmetry. First, we obtain the generating invariant operators of SO(2), i.e. $\mathcal{O}_1 = xD_y - yD_x$ and $\mathcal{O}_2 = xD_x + yD_y$. To discover this second-order PDE, we enumerate the following 5 up to second-order differential operators composed by $\mathcal{O}_1$ and $\mathcal{O}_2$: $\mathbf{O} = \{\mathcal{O}_1, \mathcal{O}_2, \mathcal{O}_1^2, \mathcal{O}_2^2, \frac{1}{x^2+y^2}(\mathcal{O}_1^2 + \mathcal{O}_2^2)\}$. Note the last operator is exactly the Laplacian. On the other hand, we have 2 fundamental differential invariants $x^2 + y^2$ and $u$. To enforce symmetry, we replace the manually defined set of extended derivatives $\{\mathcal{E}^s\}$ in D-CIPHER by all nontrivial differential functions obtained from applying an operator in $\mathbf{O}$ to one of the fundamental differential invariants. Also, instead of searching for general functions of all variables (in this case, $x, y, u$) for the RHS expression, we restrict the search space to functions of fundamental differential invariants, i.e. $x^2 + y^2$ and $u$. Since D-CIPHER uses genetic programming to find a free-form expression $g$, we can simply replace the variable set in genetic programming by $\{x^2 + y^2, u\}$ to achieve this.

Table 6: Discovery results of D-CIPHER-based methods on the Darcy flow dataset. The ground truth equation is $8(xu_x + yu_y) - (u_{xx} + u_{yy}) - e^{4(x^2+y^2)} = 0$.

| Method | Discovered equation |
|---|---|
| D-CIPHER | $xu_x + yu_y - 2.09xu_y - 2.09yu_x - 0.19u_y = 7.98x^2y^2 + 2.51xy + 0.80$ |
| D-CIPHER-SI (ours) | $(xu_x + yu_y) - 0.13(u_{xx} + u_{yy}) = 0.13e^{4.12(x^2+y^2)}$ |

Table 6 shows the equations discovered by the D-CIPHER baseline and our method. Our method can find the correct functional form of the Darcy flow equation, while D-CIPHER with the original variables and derivative operators cannot. We comment that the benefit of symmetry is even greater here for D-CIPHER than for other SR methods like SINDy, because D-CIPHER requires the user to specify both the function coefficient $a(\mathbf{x})$ and the function to be differentiated $h(\mathbf{x}, \mathbf{u})$ for an extended derivative. Such choices of functions can be largely arbitrary if no prior knowledge is available. On the other hand, our symmetry-based approach automatically selects this dictionary of differential functions.

## C.6 Samples of Discovered Equations

In Table 7, we list some randomly selected equations discovered by different methods for the Boussinesq equation (6). Some methods almost consistently discover correct/incorrect equations (i.e., have success probabilities close to 1 or 0), so we only select one sample for each. For other methods with a large variance in the discovered equations, we display two samples: a correct equation and an incorrect one.

The ground truth equation in the original variables is given in (6). The ground truth equation in the symmetry invariants is given by

$$\eta_{(0,2)} + \eta_{(0,0)}\eta_{(2,0)} + \eta_{(4,0)} + 1 = 0 \tag{50}$$

Table 7: Samples of discovered equations from the observed solution of the Boussinesq equation (6). For GP-based methods, we include results from different numbers of iterations (indicated by "$N$ its"). For transformer-based methods, we include two samples for each method because of the large variance of discovered equations from different runs.

| Method | | Equation sample(s) |
|---|---|---|
| Sparse regression | PySINDy | $u_{tt} = -1.01u_{xxxx} - 0.79uu_{xx}$ |
| | PySINDy* | $u_{tt} = -1.01u_{xxxx} - 0.99u_x^2 - 0.98uu_{xx}$ |
| | SI | $\eta_{(0,2)} = -1.00 - 1.00\eta_{(4,0)} - 1.00\eta_{(0,0)}\eta_{(2,0)}$ |
| Genetic programming | PySR (5 its) | $uu_{xx} + 1.00u_{tt} + u_{xxxx} = 0$ |
| | PySR (15 its) | $uu_{xx} + u_{tt} + u_x^2 + 1.00u_{xxxx} = 0$ |
| | SI (5 its) | $1.00\eta_{(0,0)}\eta_{(2,0)} + 1.00\eta_{(0,2)} + 1.00\eta_{(4,0)} + 1 = 0$ |
| Transformer | E2E | (1) $u_{tt} = -1.13uu_{xx} - 0.98u_{xxxx} - 0.30\|u_x\|$ 
 (2) $u_{tt} = -0.85uu_{xx} - 0.75u_x^2 - 0.99u_{xxxx}$ |
| | SI | (1) $\eta_{(0,2)} = -1.05\eta_{(0,0)}\eta_{(2,0)} - 1.00\eta_{(4,0)} - 0.96$ 
 (2) $\eta_{(0,2)} = -0.81\eta_{(0,0)}\eta_{(2,0)} - 0.40\eta_{(0,0)} - 0.98\eta_{(4,0)} - 0.90$ |

Table 8 lists the equation samples discovered from the Darcy flow dataset. The ground truth equation in original variables is given in (7), and the ground truth equation in symmetry invariants is given by

$$8\zeta_2 - \Delta u - e^{4R^2} = 0, \tag{51}$$

where $\zeta_2 = xu_x + yu_y$, $\Delta u = u_{xx} + u_{yy}$, and $R^2 = x^2 + y^2$ are among the rotational invariants used in symbolic regression.

Table 8: Samples of discovered equations for the Darcy flow dataset.

| Method | | Equation sample |
|---|---|---|
| Genetic programming | PySR | $u - 0.47x^2y^2 - 0.38e^{0.09(u_{xx}+u_{yy})} + 0.20 = 0$ |
| | SI | $\zeta_2 - 0.13\Delta u - 0.13e^{4.01R^2} = 0$ |
| Transformer | E2E | $u_{xx} = -7.43\sqrt{u^2 + 0.65u_x^2}$ |
| | SI | $\Delta u = -2.56u + 0.85\zeta_2 + 0.29$ |

Finally, Table 9 lists the equation samples discovered from the reaction-diffusion dataset. The ground truth equation in original variables is given in (8) with $d_1 = d_2 = 0.1$, and the ground truth equation in symmetry invariants is given by

$$I_t = 0.1(I_{xx} + I_{yy}) + A(1 - A)$$
$$E_t = 0.1(E_{xx} + E_{yy}) - A^2 \tag{52}$$

where $I_\mu = uu_\mu + vv_\mu$ and $E_\mu = -vu_\mu + uv_\mu$ for any multiindex $\mu$ of $t, x, y$, and $A = u^2 + v^2$.

## C.7 Prediction Errors of Discovered Equations

In Table 1, we report the prediction errors of the discovered equations on the three PDE systems. Specifically, for the Boussinesq equation and the reaction-diffusion system, we simulate the discovered PDE from an initial condition for a certain time period, e.g., $t \in [0, 20]$ for the Boussinesq equation and $t \in [0, 10]$ for the reaction-diffusion system. Then, we compare the numerical solution with the ground truth solution from the same initial condition at the end of the time period.

Table 9: Samples of discovered equations for the reaction-diffusion system dataset. Each discovered result contains two equations, since this is an evolution system with two dependent variables $u, v$.

| Method | | Equation sample |
|---|---|---|
| Sparse regression | PySINDy | $\begin{cases} u_t = 0.96u - 0.97u^3 + 1.00^3 - 0.97uv^2 + 1.00u^2v + 0.09u_{xx} + 0.09u_{yy} \\ v_t = 0.96v - 1.00u^3 - 0.97v^3 - 1.00uv^2 - 0.96u^2v + 0.09v_{xx} + 0.09v_{yy} \end{cases}$ |
| | PySINDy* | $\begin{cases} u_t = 0.21u - 0.24u^3 + 1.00v^3 - 0.23uv^2 + 0.99u^2v \\ v_t = 0.21v - 1.01u^3 - 0.24v^3 - 0.99uv^2 - 0.23u^2v \end{cases}$ |
| | SI | $\begin{cases} I_t = 0.10I_{xx} + 0.10I_{yy} + 0.96A - 0.96A^2 \\ E_t = 0.10E_{xx} + 0.10E_{yy} - 1.00A^2 \end{cases}$ |
| | SI-aligned | $\begin{cases} u_t = 0.95u - 0.96u^3 + 1.00v^3 - 0.96uv^2 + 1.00u^2v + 0.09u_{xx} + 0.09u_{yy} \\ v_t = 0.95v - 1.00u^3 - 0.96v^3 - 1.00uv^2 - 0.96u^2v + 0.09v_{xx} + 0.09v_{yy} \end{cases}$ |
| Genetic programming | PySR | $\begin{cases} u_t = 0.92v \\ v_t = -0.92u \end{cases}$ |
| | SI | $\begin{cases} I_t = 0.10I_{xx} + 0.10I_{yy} + A - 1.00A^2 \\ E_t = 0.10E_{xx} + 0.10E_{yy} - 1.00A^2 \end{cases}$ |
| Transformer | E2E | $\begin{cases} u_t = 0.89u_y \\ v_t = -0.91u \end{cases}$ |
| | SI | $\begin{cases} I_t = 0 \\ E_t = 0.50\arctan(0.45E_y - 0.31E_y/(-540.12AE_y + ...) + ...) + ... \end{cases}$ |

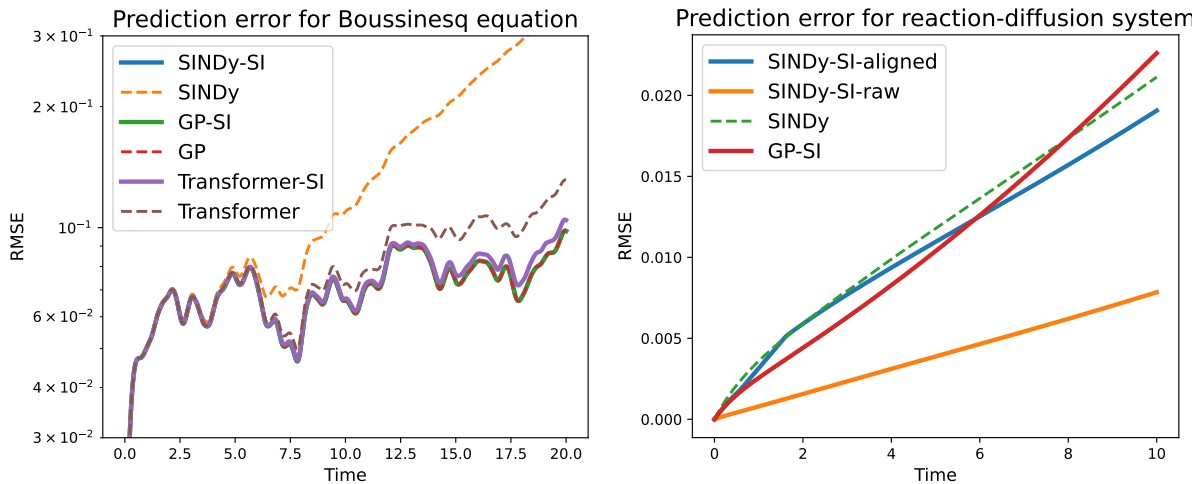

Figure 8: Prediction error over time using the discovered equations.

In addition to the prediction error at the end of the simulation time, Figure 8 shows the errors at each simulation timestep. We do not include methods whose error curves grow too fast due to the incorrectly identified equations. The results in Figure 8 are consistent with those in Table 1. Generally, the discovered equations with smaller prediction errors at the end of the simulation time also have lower prediction errors throughout the entire time interval.

For Darcy flow (7), since it describes the steady state of a system and does not involve time derivatives, we do not simulate the discovered PDEs. Instead, we evaluate each discovered PDE $F(\mathbf{x}, u^{(n)}) = 0$ on the test dataset $\{(\mathbf{x}, u^{(n)}) : \mathbf{x} \in \Omega\}$ and report the residual as the prediction error. In addition to the average error

over all the spatial grid points reported in Table 1, we visualize the error heatmaps over the grid in Figure 9. It can be observed that the discovered equations with symmetry invariants have lower errors across the entire grid.

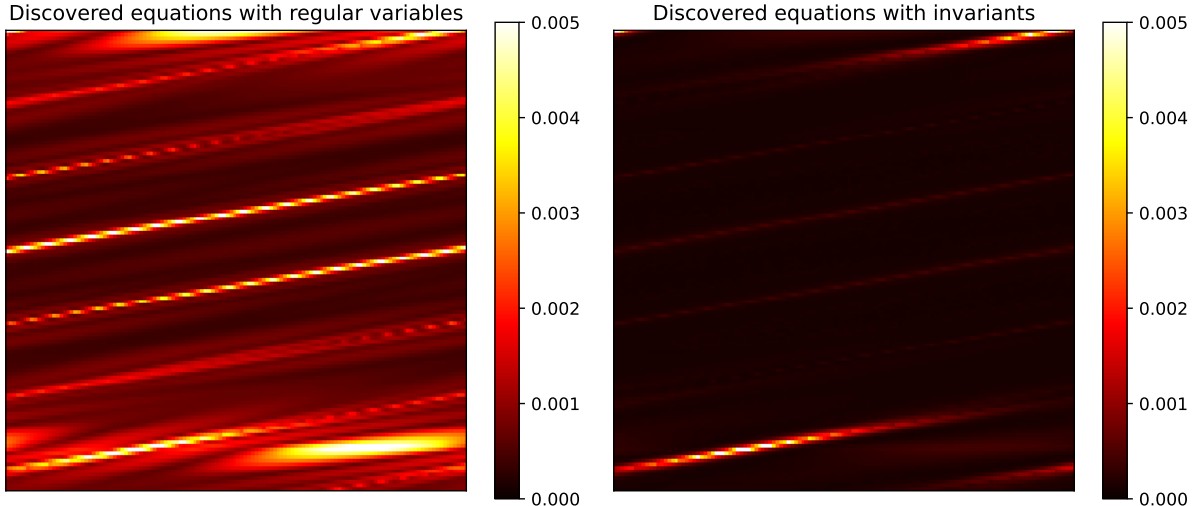

Figure 9: Prediction error of discovered equations from genetic programming methods for Darcy flow. Left: genetic programming with regular variables. Right: genetic programming with symmetry invariants.

## D   Comparison with Other Symmetry-Based Methods

In this section, we discuss the connections and differences between our work and other closely related methods that also enforce symmetry in equation discovery (Otto et al., 2023; Gurevich et al., 2024; Yang et al., 2024).

### D.1   Comparison with Methods for ODE Symmetries

Otto et al. (2023); Yang et al. (2024) focus on a special type of Lie symmetry: time-independent (TI) symmetry of ODEs. In particular, the symmetry only transforms the phase variables of ODEs. In comparison, our method can handle equations with partial derivatives and symmetries acting on the independent variables, including time and spatial variables.

When restricted to the special case of ODEs in linear combination forms and with TI symmetry, our method becomes equivalent to EquivSINDy (Yang et al., 2024). More specifically, we can still follow the procedure in Section 4.2 to construct the invariants w.r.t the specified TI symmetry, and then apply Theorem 4.4 to convert the symmetry constraint into linear constraints on the SINDy parameters. This leads to the same equivariant basis for the SINDy parameters as in EquivSINDy.

Otto et al. (2023); Yang et al. (2024) also introduce symmetry regularization, which is useful when computing the exact symmetry constraint is challenging (for example, when symmetry is learned by a neural network instead of presented in closed form). The symmetry regularization term is based on the infinitesimal criterion of Lie point symmetries, which applies to not only ODEs but also PDEs and more complex symmetries. Thus, the idea of symmetry regularization can be readily generalized to systems considered in our paper. However, as our paper primarily focuses on enforcing hard symmetry constraints, we choose not to investigate the effect of PDE symmetry regularization in great detail.

### D.2   Comparison with SPIDER: Roto-Translation Symmetry in 3D Fluid Systems

SPIDER (Gurevich et al., 2024) incorporates the symmetry of rotations and translations in 3D space into sparse regression with a weak formulation. With these specific designs, including symmetry, weak formulation,

and sparse regression, their method is shown to successfully recover several canonical equations in fluid dynamics. Notably, their method for enforcing symmetry is similar to ours: they manually construct a set of invariant scalars and a set of equivariant vectors and use them as feature sets for sparse regression.

Despite this similarity, we have established a more general framework for different types of symmetries. Examples in this paper include not only spatial rotation symmetry (Darcy flow (7)), but also scaling symmetry (Boussinesq equation (6)) and phase-space symmetry (reaction-diffusion (8)). For each of these different symmetries, we can follow the standard procedure described in Section 4 to compute their invariants. Also, in terms of the base method, SPIDER primarily focuses on the weak formulation of sparse regression. On the other hand, our method integrates with sparse regression (SINDy & WSINDy), genetic programming (PySR), and symbolic transformers.

To provide a direct experimental comparison to SPIDER (Gurevich et al., 2024), we test our method (applied to SINDy) on the same channel flow dataset specified in their Table 5 (Appendix A). The channel flow data is retrieved from the Johns Hopkins Turbulence Database (Li et al., 2008; Graham et al., 2016). The ground truth equation is given by $\mathbf{u}_t = -(\mathbf{u} \cdot \nabla)\mathbf{u} - \nabla p + \nu \Delta \mathbf{u}$, where $\nu = 5 \times 10^{-5}$.

The dataset contains a 3D velocity field $\mathbf{u}$ and a scalar pressure field $p$ over a 4D spatiotemporal grid $(x, y, z, t)$. Our method requires a set of scalar invariants of the given symmetry group. In this channel flow dataset, the assumed symmetry is rotations and spatiotemporal translations. To align with the assumption of SINDy, we specify the scalar invariant $\mathbf{u} \cdot \mathbf{u}_t$ as the LHS, and the following up-to-second-order scalar invariants as the RHS variables:

$$\mathcal{I} = \{|\mathbf{u}|^2, p, \nabla \cdot \mathbf{u}, |\boldsymbol{\omega}|^2, \mathbf{u} \cdot \boldsymbol{\omega}, \mathbf{u} \cdot [(\mathbf{u} \cdot \nabla)\mathbf{u}], \mathrm{Tr}(S^2), \mathrm{Tr}(S^3), |\nabla p|^2, \mathbf{u} \cdot \nabla p, \mathbf{u} \cdot \Delta \mathbf{u}, \Delta p, \nabla p \cdot \Delta \mathbf{u}, \boldsymbol{\omega} \cdot \Delta \mathbf{u}, |\Delta \mathbf{u}|^2\}, \quad (53)$$

where $S = \frac{1}{2}\left[\nabla \mathbf{u} + (\nabla \mathbf{u})^T\right]$ is the strain rate tensor, and $\boldsymbol{\omega} = \nabla \times \mathbf{u}$ is the vorticity. We have excluded the invariants that involve mixed derivatives (such as $\mathbf{u}_{xy}$).

Table 10: Equation discovery results of SPIDER (Gurevich et al., 2024) and our method on the channel flow dataset from the Johns Hopkins Turbulence Database.

| Method | Discovered equation |
|---|---|
| SPIDER (vector) | $1.000\mathbf{u}_t + 1.000(\mathbf{u} \cdot \nabla)\mathbf{u} + 1.000\nabla p - 0.0000500\Delta\mathbf{u} = 0$ |
| SPIDER (scalar) | $\mathbf{u} \cdot \mathbf{u}_t = -0.994\mathbf{u} \cdot (\mathbf{u} \cdot \nabla)\mathbf{u} - 1.000\mathbf{u} \cdot \nabla p + 0.0000473\mathbf{u} \cdot \Delta\mathbf{u} - 0.497\|\mathbf{u}\|^2\nabla \cdot \mathbf{u} + 0.0000069\|\nabla\mathbf{u}\|^2$ |
| SINDy-SI (ours) | $\mathbf{u} \cdot \mathbf{u}_t = 0.730\nabla \cdot \mathbf{u} - 1.008\mathbf{u} \cdot (\mathbf{u} \cdot \nabla)\mathbf{u} - 1.020\mathbf{u} \cdot \nabla p + 0.0000573\mathbf{u} \cdot \Delta\mathbf{u}$ |

Table 10 shows the discovered equations by our method (SINDy-SI) and SPIDER. The SPIDER results are referenced from their Table 1 and 2. It can be seen that SPIDER with the vector library ($\mathcal{L}_1$ in their paper) recovers the governing equation exactly. Our method and SPIDER using an extended scalar library ($\mathcal{L}'_0$ in their paper) discover mostly correct equations, with the pressure gradient, the convective term, and the Laplacian correctly identified. However, our method discovers one spurious term, the divergence $\nabla \cdot \mathbf{u}$, and SPIDER with the scalar library discovers two spurious terms ($\|\nabla u\|^2$ and $\|\mathbf{u}\|^2\nabla \cdot \mathbf{u}$). While the divergence term should be zero in theory in the incompressible flow, the numerical simulation and derivative estimation may cause a small nonzero divergence, which is then reflected in the equation discovery models.

We comment that, in this case, using the vector library in SPIDER is a natural choice, since we can easily obtain the equivariant vectors w.r.t rotations. SPIDER with the vector library also achieves the best accuracy. Still, we show that our framework, which requires an invariant scalar library, can also be applied to this scenario and identify a mostly correct equation with only one spurious term.

# E    Experiment Details

In this section, we describe the experiment setups required to reproduce the experiments.

### E.1 Data generation

**Boussinesq equation**   The equation is solved using a Fourier pseudospectral method for spatial derivatives and a fourth-order Runge-Kutta (RK4) scheme for time integration. The solution is computed on a periodic spatial domain $[-10, 10]$ with $N = 256$ grid points. The equation is reformulated as a first-order system in time by introducing $v = u_t$, and both $u$ and $v$ are evolved in time. Spatial derivatives are computed using the Fast Fourier Transform, and time derivatives of $u$ up to the fourth order are derived analytically from the governing equation. At each time step, values of $u$ are recorded in the dataset for equation discovery. The simulation starts from an initial condition of $u(x) = 0.5e^{-x^2}$ and $u_t = 0$ and proceeds up to a final time $T = 20$ with a time step of $\Delta t = 0.001$. Starting from the solution at $T = 20$, we simulate for another $T' = 20$ with the same configuration to obtain a test dataset for evaluating prediction errors of the discovered equations.

**Darcy flow**   We use the data generation code[2] from PDEBench (Takamoto et al., 2022) to generate the steady-state solution of Darcy flow over a unit square. The solution is obtained by numerically solving a temporal evolution equation

$$u_t(\mathbf{x}, t) - \nabla(a(\mathbf{x})\nabla u(\mathbf{x}, t)) = f(x), \mathbf{x} \in \Omega = (-0.5, 0.5)^2, \tag{54}$$

with $a(\mathbf{x}) = e^{-4\|\mathbf{x}\|_2^2}$, $f(\mathbf{x}) = 1$, a smooth random initial condition generated by the `init_multi_2DRand` routine from PDEBench, and homogeneous Neumann boundary conditions (zero normal flux) on $\partial\Omega$. We integrate from $t = 0$ to $t = 5$ using an explicit two-stage scheme with an adaptive time step chosen from a diffusive CFL condition, with a CFL safety factor of 0.25.

**Reaction-diffusion**   We use the data generation code[3] from PySINDy (de Silva et al., 2020; Kaptanoglu et al., 2022). We solve on a periodic spatial domain of $[-10, 10] \times [-10, 10]$ with a $128 \times 128$ Fourier spectral grid. The initial condition is given by $u(x, y, 0) = \tanh(r)\cos(\theta - r), v(x, y, 0) = \tanh(r)\sin(\theta - r)$, where $r = \sqrt{x^2 + y^2}$ and $\theta = \arg(x + iy)$. The simulation uses RK45 in Fourier space and proceeds up to a final time $T = 10$ with a time step $\Delta t = 0.05$. We perturb the numerical solution by a $0.05\%$ noise and record the values of $u, v$ to the dataset for equation discovery. Starting from the solution at $T = 10$, we simulate for another $T' = 10$ with the same configuration to obtain a test dataset for evaluating prediction errors of the discovered equations.

### E.2 Sparse regression

**Boussinesq equation**   For SINDy with original variables, we fix $u_{tt}$ as the LHS of the equation and include functions of up to 4th-order derivatives on the RHS. For PySINDy in Table 1, the library contains monomials on $U^{(4)}$ with degree in $u$ no larger than 2 and degree in any partial derivative terms $u_\alpha$ no larger than 1. For example, $u^2u_x$ is included, but $u^3$, $u_x^2$ are not. For PySINDy*, the library contains all monomials on $U^{(4)}$ up to degree 2. For example, $u_x^2$ and $uu_x$ are included. Note that the PySINDy* library does not contain all functions in the original PySINDy library, e.g., $u^2u_x$ is not included because it has degree 3.

Our method, SI, uses the invariant set in Example A.6 for sparse regression. Specifically, $\eta_{(0,2)} = u_{tt}/u_x^2$ is used as the LHS of the equation, and the rest of the invariants are included in the RHS. The function library contains all monomials of these RHS invariants up to degree 2. Also, since the invariants contain rational functions with $u_x$ on the denominator, we remove the data points with $|u_x| < 0.1$ to avoid numerical issues.

We also conduct an additional experiment to investigate the impact of the threshold value for $|u_x|$. In Table 11, we enumerate different threshold values from $\{0.0001, 0.001, 0.01, 0.1, 0.2, 0.3\}$, and report the resulting filtered dataset sizes (and their proportions compared to the unfiltered dataset), and the success probability (SP) and the prediction error (PE) metrics as in Table 1.

First of all, we notice that when the threshold value is small ($c = 0.0001$), i.e. effectively no filtering, the success probability for SINDy using invariant functions dramatically decreases. This exactly shows the

---

[2]https://github.com/pdebench/PDEBench/tree/main/pdebench/data_gen/data_gen_NLE/ReactionDiffusionEq
[3]https://github.com/dynamicslab/pysindy/blob/master/examples/10_PDEFIND_examples.ipynb

necessity of applying this numerical filter, as $u_x$ values close to zero would cause the invariant features to have large magnitudes and make the SINDy optimization unstable.

Then, as we increase $c$, we observe that our method can achieve 100% success probability for $c \in \{0.001, 0.01, 0.1\}$, showing its robustness to different choices of the threshold to some extent. When we further increase $c$, the filtered dataset becomes much smaller, and the success probability decreases. However, even with $c = 0.3$ and only 99 data points, our method is still able to recover the correct equation with more than 50% probability.

Table 11: SINDy with invariant functions on the Boussinesq equation when removing data points with $|u_x| < c$ for different threshold values $c$. In the second row, we report the number of samples in the filtered datasets and their proportions compared to the original dataset. The success probability (SP) and the prediction error (PE) are computed from 100 runs with different random seeds, in the same way as Table 1. The prediction error is reported as median [25% quantile, 75% quantile].

| Threshold $c$ | Dataset size | SP | PE |
|---|---|---|---|
| 0.0001 | 99,756 (97.4%) | 0.36 | NaN [0.129, NaN] |
| 0.001 | 97,956 (95.7%) | 1.00 | 0.103 [0.099, 0.118] |
| 0.01 | 85,591 (83.6%) | 1.00 | 0.098 [0.098, 0.099] |
| 0.1 | 26,231 (25.6%) | 1.00 | 0.098 [0.098, 0.098] |
| 0.2 | 1,318 (1.3%) | 0.91 | 0.098 [0.097, 0.108] |
| 0.3 | 99 (0.1%) | 0.52 | 0.100 [0.098, NaN] |

For all methods, we flatten the data on the spatiotemporal grid and randomly sample 2% of the data for each run. The data filtering process in SI-raw is performed after subsampling. The threshold value for sequential thresholding is set to 0.25, and the coefficient for $L_2$ regularization is set to 0.05.

**Darcy flow** Sparse regression-based methods are not directly applicable to Darcy flow (7) because there exist terms such as $e^{-4(x^2+y^2)}$. While it is still possible to include all necessary terms in the function library so that the equation can be written in the linear combination form (4), the knowledge of these complicated terms is nontrivial and should not be assumed available before running the equation discovery algorithm.

**Reaction-Diffusion** For SINDy with original variables, We fix $u_t$ and $v_t$ as the LHS of the equation and include functions of up to 2nd-order spatial derivatives on the RHS. In PySINDy, the library contains monomials of $u, v$ up to degree 3 and all spatial derivatives up to order 2. In PySINDy$^*$, the library contains all monomials of $u, v$ and their up to second-order spatial derivatives up to degree 3.

Our method uses the invariant set $\{t, x, y, u^2 + v^2\} \bigcup \{\mathbf{u} \cdot \mathbf{u}_\mu\} \bigcup \{\mathbf{u}^\perp \cdot \mathbf{u}_\mu\}$, where $\mathbf{u} = (u, v)^T$ and $\mu$ is a multiindex of $t, x, y$. We will denote $I_\mu = \mathbf{u} \cdot \mathbf{u}_\mu$ and $E_\mu = \mathbf{u}^\perp \cdot \mathbf{u}_\mu$. We use $I_t$ and $E_t$ as the LHS of the equation, and the rest of the invariants are included in the RHS. The function library contains all monomials of these RHS invariants up to degree 2.

We randomly sample 10% of the data for each run. The threshold value for sequential thresholding is set to 0.05. The coefficient for $L_2$ regularization is set to 0 for SINDy with original variables and 0.1 for our method with symmetry invariants.

For the experiments with different levels of noise (Section 5.4), we use weak SINDy as the base algorithm. We use the implementation of weak SINDy from the PySINDy package (Kaptanoglu et al., 2022). The function library is the same as SINDy as described above. To enforce symmetry, instead of directly using the symmetry invariants, we derive a set of linear constraints on the sparse regression parameters to adapt to weak SINDy. This procedure is further described in Appendix B.3.

### E.3 Genetic Programming

In all experiments, to determine if an equation matches the ground truth we first expand the prediction into a sum of monomial terms. We then eliminate all terms whose relative coefficient is below 0.01. For each

term in the filtered expression, we see if it matches any term in the ground truth expression. This is done by randomly sampling 100 points from the standard normal distribution and evaluating both the prediction and candidate ground truth term on the generated points. Note that we drop the coefficients before evaluation. If all evaluations of the predicted term have a relative error of less than 5% from those of the ground truth, the terms are said to match. If there is a perfect matching between the terms in the ground truth and prediction, the prediction is listed as correct.

Rather than directly returning a single equation, PySR finally produces a hall-of-fame that consists of multiple candidate solutions with varying complexities. To finally pick a single prediction, we use a selection strategy equivalent to the "best" option from PySR.

**Boussinesq equation** For the Boussinesq equation (6), we first randomly subsample 10000 datapoints. We configure PySR to use the addition and multiplication operators, to have 127 populations of size 27, and to have the default fraction-replaced coefficient of 0.00036.

When running with ordinary variables, we sequentially try fixing the LHS to each variable in $(\mathbf{x}, u^{(4)})$ and allow the RHS to be a function of all remaining variables. Similarly, runs using invariants sequentially fix the LHS from the set given by Example A.6 and the RHS as a function of all other invariants.

For each iteration count of 5, 10, and 15, we run the algorithm using invariant or ordinary variables and report the number of correct predictions out of 100 trials.

**Darcy flow** In the Darcy experiment (7), we eliminate all points that are within 3 pixels from the border and then randomly subsample 10000 datapoints. We configure PySR to use the addition, multiplication, and exponential operators; to have 127 populations of size 64; and to have a fraction-replaced coefficient of 0.1. We further constrain it to disallow nested exponentials (e.g. $\exp(\exp(x) + 4)$).

We try all possible ordinary variables in $(\mathbf{x}, u^{(2)})$ for the LHS and the RHS is then a function of the unused variables. Likewise when using invariants, we fix the LHS to each possible invariant specified in Example A.5 and set the RHS as a function of the remaining invariants.

For each iteration count of 50, 100, and 200, we run the algorithm using invariant or ordinary variables and report the number of correct predictions out of 100 trials.

**Reaction-Diffusion** For the Reaction Diffusion equation (8), we remove all points that are within 3 pixels from the border or have timestamp greater than or equal to 40, and then randomly subsample 10000 datapoints. We configure PySR to use the addition and multiplication operators, to have 127 populations of size 64, and to have a fraction-replaced coefficient of 0.5.

In the ordinary variable case, we fix the LHS as either $u_{tt}$ or $v_{tt}$ and allow the RHS to be a function of all other variables in $(\mathbf{x}, u^{(2)})$. When using invariants, the LHS is fixed to be either $I_t$ or $E_t$ and the RHS is then a function of all remaining invariants.

For each iteration count of 100, 200, and 400, we run the algorithm using regular and ordinary variables and report the number of correct predictions out of 100 trials.

### E.4 Symbolic Transformer

We use the pretrained symbolic transformer model provided in the official codebase[4] from Kamienny et al. (2022). The transformer-based symbolic regressor is initialized with 200 maximal input points and 100 expression trees to refine. The variable sets used in the symbolic transformer are the same as those described in the genetic programming experiments, except for the Boussinesq equation, where we remove all mixed derivative terms in both the original variable set and the symmetry invariant set. We find that the symbolic transformer can sometimes discover the correct equation under this further simplified setup, but fails when using the larger variable sets.

---

[4]https://github.com/facebookresearch/symbolicregression/blob/main/Example.ipynb

We also fix the LHS of the function and use the remaining variables as RHS features. For the Boussinesq equation, the LHS is fixed to $u_{tt}$ for original variables and $\eta_{(0,2)}$ for symmetry invariants. For the Darcy flow, the LHS is fixed to $u_{xx}$ for original variables and $\Delta u$ for symmetry invariants. For the reaction-diffusion system, the LHS is fixed to $u_t, v_t$ for original variables and $I_t, E_t$ for symmetry invariants.

### E.5 Hypothesis Spaces of Equation Discovery Algorithms

Table 12 and Table 13 describe the hypothesis spaces of different equation discovery algorithms when applied to the Boussinesq equation and the reaction-diffusion system.

Table 12: Hypothesis spaces of different equation discovery algorithms for the Boussinesq equation.

| Method | | Hypothesis space |
|---|---|---|
| Sparse Regression | PySINDy | $u_{tt} = W\boldsymbol{\theta}(u^{(4)}), \{\theta^j\} = \{ab : a \in \mathrm{Mono}_{\leq 2}(U), b \in \{1, u_x, ..., u_{xxxx}\}\}$ |
| | PySINDy$^*$ | $u_{tt} = P(u^{(4)}) \in \mathrm{Poly}_{\leq 2}(U^{(4)})$ |
| | PySINDy$^{**}$ | $u_{tt} = W\boldsymbol{\theta}(u^{(4)}), \{\theta^j\} = \{u^{c_0}u_1^{c_1}u_2^{c_2}u_3^{c_3}u_4^{c_4} : c_i \geq 0, \sum_0^4 c_i \leq 2, \sum_1^4 \mathrm{sgn}(c_i) \leq 1\}$ |
| | SI | $\eta_{(0,2)} = P(\boldsymbol{\eta}) \in \mathrm{Poly}_{\leq 2}(\{\eta_{(\alpha,\beta)}\}\backslash\{\eta_{(0,2)}\})$ |
| Genetic Programming | PySR | $z^j = f(\mathbf{z}^{-j})$ for $\mathbf{z} = (\mathbf{x}, u^{(4)})$ and some $j$ |
| | SI | $\eta_{(\alpha_0,\beta_0)} = f(\boldsymbol{\eta}_{-(\alpha_0,\beta_0)})$ for $\boldsymbol{\eta} = \{\eta_{(\alpha,\beta)} : \alpha + \beta \leq 4\}$ and some $(\alpha_0, \beta_0)$ |

Table 13: Hypothesis spaces of different equation discovery algorithms for 2D reaction-diffusion. $\mathbf{u}^{(n)} \in U^{(n)}$ denotes the collection of all up to $n$th order *spatial* derivatives. $\alpha = [\alpha_1, \alpha_2]$ is the multiindex for spatial variables. $\mathbf{x} = (x, y, t)$. $A = u^2 + v^2$.

| Method | | Hypothesis space |
|---|---|---|
| Sparse Regression | PySINDy | $\mathbf{u}_t = W\boldsymbol{\theta}(\mathbf{u}^{(2)}), \{\theta^j\} = \mathrm{Mono}_{\leq 3}(U) \bigcup \{\mathbf{u}_\alpha : |\alpha| \leq 2\}$ |
| | PySINDy$^*$ | $\mathbf{u}_t = P(\mathbf{u}^{(2)}) \in \mathrm{Poly}_{\leq 3}(U^{(2)})$ |
| | PySINDy$^{**}$ | $\mathbf{u}_t = W\boldsymbol{\theta}(\mathbf{u}^{(2)}), \{\theta^j\} = \{\prod_{i,\alpha}(u_\alpha^i)^{c_\alpha^i} : \sum c_\alpha^i \leq 3, \sum_{|\alpha| \geq 1}\mathrm{sgn}(c_\alpha^i) \leq 1\}$ |
| | SI | $[I_t, E_t]^T = P \in \mathrm{Poly}_{\leq 2}(A, \mathbf{x}, I_\alpha, E_\alpha; |\alpha| \leq 2)$ |
| | SI-aligned | $\mathbf{u}_t = W\boldsymbol{\theta}(\mathbf{u}^{(2)}), W^{jk} = Q^{i\bar{j}\bar{k}}\beta^i$ for some precomputed $Q$ |
| Genetic Programming | PySR | $\mathbf{u}_t = \mathbf{f}(\mathbf{x}, \mathbf{u}^{(2)})$ |
| | SI | $[I_t, E_t]^T = \mathbf{f}(A, \mathbf{x}, I_\alpha, E_\alpha; |\alpha| \leq 2)$ |

