# OpenReview forum: "Discovering Symbolic Differential Equations with Symmetry Invariants"
_TMLR — Accepted by TMLR_

### Review · Reviewer_FLuz · 2026-01-17

**Summary Of Contributions:**

This paper introduces a general framework for enforcing symmetry constraints in symbolic regression (SR) methods used for discovering partial differential equations (PDEs). The key idea is to replace the standard set of variables and derivatives with differential invariants of a given symmetry group. This ensures that any equation discovered by the SR algorithm inherently respects the specified symmetry, thereby reducing the search space, improving parsimony, and preventing unphysical solutions.

Strenghts:
-The method is derived from the well-established theory of differential invariants and can be integrated with various existing SR backbones (sparse regression, genetic programming, transformers).
-The paper provides comprehensive experiments on canonical PDE systems (Boussinesq, Darcy flow, reaction-diffusion), demonstrating improvements in accuracy, efficiency, and robustness—especially in challenging scenarios with noisy data and imperfect symmetry.
-The proposed relaxation of strict symmetry constraints (Section 4.4) is a pragmatic solution for handling systems with approximate or broken symmetries, showing better performance than an unconstrained baseline.

Weaknesses:
-As already noted by the authors, the method assumes the symmetry group is known a priori. The framework cannot discover symmetry itself and may produce incorrect results if the symmetry is misspecified.
-While the literature review is excellent within the specific scope of symbolic regression, its impact could be enhanced by briefly situating the work within the broader, compelling challenge of learning or discovering governing equations from data.

**Audience:**

Yes

**Audience Explanation:**

Yes. The paper’s integration of symmetry theory with scalable symbolic regression directly addresses a core interest in interpretable PDE discovery, a topic relevant to researchers in machine learning, scientific computing, and computational physics within the TMLR audience.

**Broader Impact Concerns:**

No broader impact concerns. The authors have included a clear and appropriate broader impact statement, acknowledging both the potential scientific benefits and the essential need for domain-expert validation to mitigate risks from erroneous equations. The work presents no identifiable or specific ethical risks.

**Claims And Evidence:**

Yes

**Claims Explanation:**

The claims made in the submission are supported by accurate, convincing, and clear evidence, as demonstrated through a multi-faceted validation strategy:

-The core claim, that symmetric PDEs can be expressed via differential invariants, is anchored in established mathematical theory. The paper correctly leverages Theorem 4.2 as its foundation and provides derivations for constructing invariants.
-The method is tested on three distinct, canonical PDE systems (Boussinesq, Darcy flow, reaction-diffusion) covering different challenges (high-order derivatives, multiple variables, steady-state vs. evolution).
-Performance is compared fairly against standard implementations of three major SR algorithm classes (SINDy, PySR, E2E transformer) both with and without the symmetry invariant (SI) framework.
-Success is measured using clear, relevant metrics: Success Probability (SP) for recovery of the exact symbolic form, and Prediction Error (PE) for the predictive fidelity of the discovered model.
-Experiments systematically vary key conditions to test robustness: e.g. Noisy Data or Imperfect Symmetry
-The integration of SI into different SR backbones is explained with explicit algorithms.
- Implementation details, data generation procedures, and hyperparameters are thoroughly documented in the Appendices.
-The paper acknowledges limitations openly.

**Requested Changes:**

The paper is a well-executed and significant contribution that merits publication. My recommendation for acceptance is not contingent on any changes. However, the following minor suggestion would strengthen the work by providing a more comprehensive context for the broader TMLR audience.

Minor Suggestion to Strengthen the Work:
The current related work section focuses primarily on symbolic regression methods. To better position this contribution within the extensive field of PDE discovery and learning from data, please consider adding a brief paragraph or subsection that acknowledges other major paradigms. This could include, but is not limited to, neural operator learning (e.g., DeepONet), Neural Networks and Gaussian process-based PDE learning, and methods for learning effective coarse-grained dynamics from fine-scale simulations. This addition would help readers understand how this symbolic, symmetry-informed approach complements and differs from non-symbolic or surrogate modeling techniques in the long-standing pursuit of discovering governing equations from data.

The current related work section focuses primarily on symbolic regression methods. To better position this contribution within the extensive field of PDE discovery and learning from data, I respectfully suggest the authors consider adding a brief paragraph or subsection that acknowledges other major paradigms. This could include neural operator learning, Gaussian process-based methods, and techniques for learning effective coarse-grained dynamics or closures from fine-scale simulations (a long-standing goal in numerical analysis and scientific machine learning).

The authors may of course select their own preferred set of references to represent these directions. For their consideration, the following works exemplify the breadth of related approaches:


Lee et al. (2020). Coarse-scale PDEs from fine-scale observations via machine learning. Chaos.

Vlachas et al. (2020). Backpropagation algorithms and reservoir computing for spatiotemporal dynamics. Neural Networks.

Li et al. (2020). Hamiltonian neural networks. Proceedings of the 2nd Symposium on Advances in Approximate Bayesian Inference.

Chen et al. (2021). Solving and learning nonlinear PDEs with Gaussian processes. Journal of Computational Physics.

Lu et al. (2021). Learning nonlinear operators via DeepONet. Nature Machine Intelligence.

Galaris et al. (2022). Numerical bifurcation analysis of PDEs from lattice Boltzmann model simulations. Journal of Scientific Computing.

Vlachas et al. (2022). Multiscale simulations of complex systems by learning their effective dynamics. Nature Machine Intelligence.

Floryan & Graham (2022). Data-driven discovery of intrinsic dynamics. Nature Machine Intelligence.

Lee et al. (2023). Learning chemotactic PDEs/closures from agent-based data. Journal of Mathematical Biology.

Dietrich et al. (2023). Learning effective stochastic differential equations. Chaos.

Fabiani et al. (2024). Task-oriented machine learning surrogates for tipping points of agent-based models. Nature Communications.

A concise discussion situating the current symmetry-informed symbolic approach alongside such non-symbolic or surrogate modeling techniques would be valuable for readers.

---

> ### Author Response · Authors · 2026-02-14
>
> We thank the reviewer for the valuable feedback and the detailed list of references. We have updated Section 2 to also include a paragraph about PDE learning and surrogate modeling with the mentioned references.

---

### Review · Reviewer_YyRe · 2026-01-19

**Summary Of Contributions:**

The authors present a novel approach for symbolic discovery of differential equations by using symmetry invariant groups as the atomic entities in equation discovery. The approach is sophisticated but well-explained and involves learning a number of low-order invariants then composing those to construct sets of arbitrary orders. The paper is extremely well-written. The authors then show the behavior of their method on three PDE examples. The new method performs favorably to other methods such as SINDy and genetic programming.

**Audience:**

Yes

**Audience Explanation:**

This work is very timely in the field of scientific machine learning. Discovering equations from data in a fashion that respects underlying group structure will be critical for both trustworthy surrogate modeling and for learning in data-poor regimes.

**Broader Impact Concerns:**

None.

**Claims And Evidence:**

Yes

**Claims Explanation:**

The authors present results on three PDEs, and even explore the impact of noise. The results are convincing. That said, there are some issues (see below).

**Requested Changes:**

I would recommend the following changes. These are crucial both to strengthening the work and for acceptance.

1. For the PDEs that are presented, the authors should provide a complete description in the main body of the paper. It should include the domain in which the PDE is solved, the dimension, and the and the boundary conditions. Without this information, it's almost impossible to replicate results. That said, it looks like all the PDEs are in two spatial dimensions at most.

2. I'd have liked to see both 3D and higher dimensional PDE examples. It's in these scenarios that discovery of symmetries will be particularly important, and these are the scenarios with greatest practical applicability.

3. It is important to see nonlinear examples as well.

4. Could the authors show with one example how the method performs when third or higher order invariants are involved? Perhaps an ablation over the order of invariants, if that's at all possible? Given enough solution smoothness, that should be doable in principle, though there may be practical difficulties.

---

> ### Author Response · Authors · 2026-02-14
>
> We thank the reviewer for the valuable feedback. We address the requested changes below:
>
> > For the PDEs that are presented, the authors should provide a complete description in the main body of the paper. It should include the domain in which the PDE is solved, the dimension, and the and the boundary conditions. Without this information, it's almost impossible to replicate results.
>
> We have included these descriptions in Appendix D.1.
>
> > I'd have liked to see both 3D and higher dimensional PDE examples.
>
> In Appendix C.4, we have added an example of discovering a reaction-diffusion equation in 3 spatial dimensions.
>
> Also, in response to Reviewer hkwD, we have added a comparison with the SPIDER paper (Gurevich et al., 2024) in Appendix D.2, which includes an experiment of discovering the 3D incompressible Navier-Stokes equation as another higher-dimensional example.
>
> > It is important to see nonlinear examples as well.
>
> We agree that nonlinear PDEs are important. In fact, two of our three benchmark equations (Boussinesq and the reaction–diffusion system) are already nonlinear. The Darcy flow example serves as a linear case.
>
> > Could the authors show with one example how the method performs when third or higher order invariants are involved?
>
> If the reviewer means how our method performs on *ground truth equations* with higher-order invariants, we have presented in the paper the Boussinesq equation (7) as a fourth-order example.
>
> On the other hand, if the question is how the method performs when the ground truth equation only involves first- or second-order invariants, but we choose a feature library that includes higher-order invariants, we provide an ablation below on the reaction-diffusion system (8). We include the third- and fourth-order invariants in the feature sets of sparse regression and genetic programming, while keeping other setups the same as in Table 1 in the paper.
>
> | Method   | $n=2$ | $n=3$ | $n=4$ |
> |----------|-------|-------|-------|
> | PySINDy  | 0.53  | 0.52  | 0.52  |
> | SINDy-SI | 0.54  | 0.54  | 0.54  |
> | PySR     | 0.00  | 0.00  | 0.00  |
> | PySR-SI  | 0.81  | 0.00  | 0.00  |
>
> As the table above shows, sparse regression methods, including the baseline PySINDy and our SINDy-SI, are robust when the irrelevant higher-order invariants are included in the library. The sequential thresholding procedure in SINDy filters out these additional terms effectively in the first few iterations. On the other hand, genetic programming (GP) methods do not work well when higher-order invariants are added. This is likely because the addition of these invariants makes the number of features too large for PySR to handle. Since GP explores the entire equation space by random mutations, a larger feature set would prevent GP from identifying any plausible expression with only relevant variables within the restricted computation budget.

---

### Review · Reviewer_hkwD · 2026-02-08

**Summary Of Contributions:**

This article proposes a regression-based equation learning framework that relies on differential invariants of symmetry transformations as the fundamental building blocks, rather than raw variables. The goal of doing so is to ensure that the discovered equation satisfies existing physical symmetries.

This article is clear, well-written, and technically correct.  Most of the comments below are designed to improve the impat ofthe paper.

**Audience:**

Yes

**Audience Explanation:**

One of the major challenges of using equation error-based equation learning is that it is far too easy for non-physical terms to appear, and then the discovered model is completely wrong.  This article addresses that lack head on.

**Broader Impact Concerns:**

The broader impact is fine. I have no comments.

**Claims And Evidence:**

Yes

**Claims Explanation:**

The authors propose their methodology and describe it in great detail. The efficacy of their proposed method is demonstrated on a handful of canonical examples. They do present some theory in the appendix, but this one lemma  is focused on the differential invariant symmetry transformations as the fundamental building blocks, rather than raw variables. The second does provide some comfort regarding convergence. However, convergence is different when using the strong vs. weak forms of SINDy, so it's not clear how translatable the theory is.

In support of the theory, then demonstrate successful learning of the equations for several PDE examples (even in the presence of noise).

**Requested Changes:**

* Figure 3 SINDy -> WSINDy
* Are the authors using WSINDy or the weak form implementation in pysindy. As WSINDy autotunes its internal parameters, it can be directly applied to data. The weak form implemented in pysindy is not the same as WSINDy and does not autotune parameters (like the test function radius).  There are several examples of Pysindy failing where WSINDy succeeded.
* Given the strong similarity to the SPIDER paper by Gurevich et al., 2024 in JFM, the authors must compare with the results of that code. Of course, either the SPIDER code or the authors' code must be altered to be comparable.  The goals and overall approach of the two methods are just so similar that the authors need to include a serious discussion of the similarities and differences.
* While there are several examples of implementing the proposed framework to enforce symmetries of differential equations. I would like to see a discussion on how easy it is to adapt an algorithm with the symmetry constraints described. They are described in technical detail, but a qualitative review would be helpful. Perhaps some of the material in Appendix C.3 can be summarized in 5.3.
* In the paper, the authors state, “For more general symmetries and physical systems, enforcing symmetry often requires additional assumptions on the form of equations, such as the linear combination form in sparse regression (Otto et al., 2023; Yang et al., 2024).” One of my questions is how the proposed work compares to this method when an ODE can be written as a linear combination. Perhaps in general, it would be good to make a comparison to the other methods the authors reference that are able to enforce symmetry.

---

> ### Author Response · Authors · 2026-02-14
> **Response, Part I**
>
> We thank the reviewer for the valuable feedback. We address the requested changes below.
>
> > Are the authors using WSINDy or the weak form implementation in pysindy. As WSINDy autotunes its internal parameters, it can be directly applied to data. The weak form implemented in pysindy is not the same as WSINDy and does not autotune parameters (like the test function radius). There are several examples of Pysindy failing where WSINDy succeeded.
>
> We thank the reviewer for pointing this out. We were using PySINDy’s weak form implementation in the original manuscript. In the revised version, we have updated Figure 4 (left), including both autotuned parameters following the procedure in WSINDy (Messenger & Bortz, 2021) and the default parameters specified in PySINDy. Specifically, we implemented their [findcorners subroutine](https://github.com/MathBioCU/WSINDy_PDE/blob/master/findcorners.m) in Python and used its results to instantiate the PySINDy WeakPDELibrary with corresponding test function parameters. It is shown in the figure that the autotuned WSINDy parameters increase the success probability in most cases.
>
> We have kept Figure 4 (center & right) for the RD systems with imperfect symmetry the same (with default PySINDy parameters). In Appendix C.3, we include another figure for experiments on these systems, using the same setup except for auto-tuned WSINDy parameters.
>
> > Given the strong similarity to the SPIDER paper by Gurevich et al., 2024 in JFM, the authors must compare with the results of that code. Of course, either the SPIDER code or the authors' code must be altered to be comparable. The goals and overall approach of the two methods are just so similar that the authors need to include a serious discussion of the similarities and differences.
>
> We recognize that both the SPIDER paper and our manuscript share a common goal of discovering PDEs from data using symmetry as a constraint. However, their paper focuses on recovering equations for 3D fluid systems, which admit a **specific** symmetry group of spatiotemporal translations and $\mathrm O(3)$ rotations/reflections. In comparison, we establish a general framework for different types of symmetries. Examples in our manuscript include not only spatial rotation symmetry (Darcy flow), but also scaling symmetry (Boussinesq equation) and phase-space symmetry (reaction-diffusion).
>
> Also, in terms of the **base method**, SPIDER primarily focuses on the weak formulation of SINDy-PI and uses symmetry and other techniques to improve upon it. On the other hand, our method integrates with sparse regression (SINDy & WSINDy), genetic programming (PySR), and symbolic transformers. We indeed utilize the weak formulations, but present them as one possible implementation of our framework.
>
> To compare with SPIDER, we test our method (applied to SINDy) on the same channel flow dataset specified in their Table 5 (Appendix A). Our method requires a set of scalar invariants of the given symmetry group. In this channel flow dataset, the assumed symmetry is rotations and spatiotemporal translations. To align with the assumption of SINDy, we specify the scalar invariant $\mathbf u \cdot \mathbf u_t$ as the LHS, and 15 up-to-second-order scalar invariants as the RHS variables. The discovered equations are shown in the table below. The SPIDER results are referenced from their Table 1 and 2.
>
> | Method | Discovered equation |
> |-|-|
> | SINDy-SI (ours) | $\mathbf u \cdot \mathbf u_t = 0.730\nabla \cdot u - 1.008\mathbf u\cdot (\mathbf u\cdot\nabla)\mathbf u-1.020\mathbf u\cdot\nabla p + 0.0000573\mathbf u\cdot\Delta\mathbf u$ |
> | SPIDER (vector) | $1.000 \mathbf u_t + 1.000 (\mathbf u\cdot\nabla)\mathbf u + 1.000 \nabla p - 0.0000500 \Delta u = 0$ |
> | SPIDER (scalar) | $\mathbf u \cdot \mathbf u_t = -0.994 \mathbf u\cdot (\mathbf u\cdot\nabla)\mathbf u - 1.000\mathbf u\cdot\nabla p + 0.0000473\mathbf u\cdot\Delta\mathbf u - 0.497 \|\mathbf u\|^2\nabla\cdot\mathbf u + 0.0000069 \|\nabla u\|^2$ |
>
> It can be seen that SPIDER with the vector library (eq. (3) in their paper) recovers the governing equation exactly. Our method (SINDy-SI) and SPIDER using an extended scalar library (eq. (15) in their paper) discover mostly correct equations, with the pressure gradient, the convective term and the Laplacian correctly identified. However, our method discovers one spurious term, the divergence $\nabla \cdot \mathbf u$, and SPIDER with the scalar library discovers two spurious terms ($\|\nabla u\|^2$ and $\|\mathbf u\|^2\nabla\cdot\mathbf u$). While the divergence term should be zero in theory in the incompressible flow, the numerical simulation and derivative estimation may cause a small nonzero divergence, which is reflected in the equation discovery models.

---

> ### Author Response · Authors · 2026-02-14
> **Response, Part II**
>
> (continued) We comment that, in this case, using the vector library in SPIDER is a natural choice, since we can easily obtain the equivariant vectors w.r.t rotations. SPIDER with the vector library also achieves the best accuracy. Still, we show that our framework, which requires an invariant scalar library, can also be applied to this scenario and identify a mostly correct equation with only one spurious term.
>
> > While there are several examples of implementing the proposed framework to enforce symmetries of differential equations. I would like to see a discussion on how easy it is to adapt an algorithm with the symmetry constraints described. They are described in technical detail, but a qualitative review would be helpful. Perhaps some of the material in Appendix C.3 can be summarized in 5.3.
>
> The complexity of adapting our symmetry-based framework lies not in the generation of invariants, but in aligning them with the structural assumptions of the base algorithm. Conceptually, the adaptation aims to find the “intersection of hypothesis spaces” as we show in the Venn diagram in Figure 2. In general, if the base algorithm has a more nontrivial and structured hypothesis about possible equation forms, then more effort is required to apply our method to it.
>
> For instance, integrating our method with genetic programming/symbolic transformers is a direct feature substitution, while integrating with SINDy requires aligning the invariant library with the algorithm's specific linear-combination architecture and LHS/RHS partitioning. Moreover, integrating with the D-CIPHER base algorithm as described in the original Appendix C.3 (now C.5) requires more effort, because D-CIPHER’s hypothesis space of equations is formulated by a combination of extended derivatives and free-form expressions, which we need to handle separately.
>
> We have modified the beginning of Sec 4.3 to include a qualitative statement regarding the reviewer’s point.
>
> > In the paper, the authors state, “For more general symmetries and physical systems, enforcing symmetry often requires additional assumptions on the form of equations, such as the linear combination form in sparse regression (Otto et al., 2023; Yang et al., 2024).” One of my questions is how the proposed work compares to this method when an ODE can be written as a linear combination. Perhaps in general, it would be good to make a comparison to the other methods the authors reference that are able to enforce symmetry.
>
> The mentioned papers (Otto et al., 2023; Yang et al., 2024) focus on a special type of Lie symmetry: time-independent (TI) symmetry of ODEs. In particular, the symmetry only transforms the phase variables of ODEs. In comparison, our method can handle equations with partial derivatives and symmetries acting on the independent variables, including time and spatial variables.
>
> When restricted to the special case of ODEs in linear combination forms and with TI symmetry, our method will become equivalent to EquivSINDy (Yang et al., 2024). More specifically, we can still follow the procedure in Sec 4.1 & 4.2 to construct the invariants w.r.t the specified TI symmetry, and then apply Prop 4.4 to convert the symmetry constraint into linear constraints on the SINDy parameters. This leads to the same equivariant basis for the SINDy parameters as in EquivSINDy.
>
> Otto et al. (2023) and Yang et al. (2024) also introduced symmetry regularization, which is useful when computing the exact symmetry constraint is challenging (for example, when symmetry is learned by a neural network instead of presented in closed form). The symmetry regularization term is based on the infinitesimal criterion of Lie point symmetries, which applies to not only ODEs but also PDEs and more complex symmetries. Thus, the idea of symmetry regularization can be readily generalized to systems considered in our paper. However, as our paper primarily focuses on enforcing hard symmetry constraints, we choose not to investigate the effect of PDE symmetry regularization in great detail.
>
> In the revision, we have added a reference to Appendix D at the end of Sec 2 (Related Work), which compares our method with other methods that use symmetry. This includes the responses to this point and the previous point about SPIDER.

---

### Decision · Action_Editor_pNZc · 2026-03-03

**Recommendation:** Accept as is

**Additional Comments:**

2 of the 3 reviewers spoke especially highly of the quality of this work, with one recommending Featured Certification. Looking through the submission, I am in agreement and also recommend for Featured Certification.

All 3 reviewers had their comments addressed, therefore I am recommending accept as is.

**Audience:**

Yes

**Audience Explanation:**

All 3 reviewers agreed that the work would be of interest to the broader TMLR community.

**Claims And Evidence:**

Yes

**Claims Explanation:**

All 3 reviewers agreed that the claims were supported by accurate, convincing, and clear evidence.